# Improving regional air quality predictions in the Indo-Gangetic Plain - Case study of an intensive pollution episode in November 2017.

Behrooz Roozitalab[1,2], Gregory R. Carmichael[1,2], Sarath K. Guttikunda[3]

[1]Center for Global and Regional Environmental Research, University of Iowa, Iowa City, IA, USA,
[2]Chemical and Biochemical Engineering, University of Iowa, Iowa City, IA, USA
[3]Urban Emissions, New Delhi, India

*Correspondence to:* B. Roozitalab (behrooz-roozitalab@uiowa.edu) and G. R. Carmichael (gcarmich@engineering.uiowa.edu)

**Abstract.** Indo-Gangetic Plain (IGP) experienced an intensive air pollution episode during November 2017. Weather Research and Forecasting model with Chemistry (WRF-Chem), a coupled meteorology–chemistry model, was used to simulate this episode. In order to capture $PM_{2.5}$ peaks, we modified input chemical boundary conditions and biomass burning emissions. CAM-Chem and MERRA-2 global models provided gaseous and aerosol chemical boundary conditions, respectively. We also incorporated VIIRS active fire points to fill missing fire emissions in FINN and scaled by a factor of seven for an 8-days period. Evaluations against various observations indicated the model captured the temporal trend very well although missed the peaks on Nov. 7[th], 8[th], and 10[th]. Modeled aerosol composition in Delhi showed Secondary Inorganic Aerosols (SIA) and Secondary Organic Aerosols (SOA) comprised 30% and 27% of total $PM_{2.5}$ concentration, respectively, during November, with a modeled OC/BC ratio of 2.72. Back trajectories showed agricultural fires in Punjab were the major source for extremely polluted days in Delhi. Furthermore, high concentrations above the boundary layers in vertical profiles suggested either the plume rise in the model released the emissions too high, or the model did not mix the smoke down fast enough. Results also showed long-range transported dusts did not affect Delhi's air quality during the episode. Spatial plots showed averaged Aerosol Optical Depth (AOD) of 0.58 (±0.4) over November. The model AODs were biased high over central India and low over eastern IGP, indicating improving emissions in eastern IGP can significantly improve the air quality predictions. We also found high ozone concentrations over the domain, which indicates ozone should be considered in future air quality management strategies alongside particulate matters.

## 1. Introduction

Ambient air pollution remains a major environmental issue, even after significant worldwide efforts starting after the deadly smog of London in 1952. It is the fifth-ranking risk of death and a major threat to climate and ecosystem (Cohen et al., 2017;Ramanathan and Carmichael, 2008;Sitch et al., 2007). Air pollution contains many species; particulate matter (PM) is currently the air pollutant of most concern, especially in developing countries like India. India is an emerging economy with burgeoning population that has accelerated its industrial activities in the last three decades, leading to wide spread air pollution and resulting adverse health effects. There are many Indian cities on the list of most polluted cities of the world (World-Bank, 2018;Guttikunda et al., 2014;WHO, 2016). Studies show that ozone and particulate matter with diameter less than 2.5 micron ($PM_{2.5}$), are attributed to more than one million individual premature deaths in India (Cohen et al., 2017;HEI, 2018).David et al. (2019) found that anthropogenic emissions within India led to about 80% of the total premature death due to $PM_{2.5}$ in India. Furthermore as industrial activities are growing, emissions are increasing too; health impacts attributed to long-term exposure air pollution are predicted to increase based on current policies (Conibear et al., 2018a).

Short-term extreme pollution events lead to increased hospital admissions and mortalities (Anenberg et al., 2018;Rajak and Chattopadhyay, 2019). Forest and agricultural fires, dust storms, increased local activities, and stagnant meteorological conditions

are major contributing factors in these air pollutions episodes (Beig et al., 2019;Jethva et al., 2018). While forecasting models help authorities to notify people of these extreme pollution events, hindcasting models help scientists improve the capabilities of the models to predict pollution events, identify the main responsible factors causing these events, and inform policy makers as they develop pollution control strategies. However, the ability of air quality models for simulating short-term events highly depends on the quality of input chemical data (i.e. emissions). For example, the total amount of global fire emissions can differ by a factor of 3-4 based on the emission inventory used (Pan et al., 2020). Furthermore, dust storms can travel long distances and influence another region's air quality (Ashrafi et al., 2017;Beig et al., 2019). David et al. (2019) attributed about 16% of total premature $PM_{2.5}$-related death to emissions outside India. Moreover, studies of Black Carbon (BC) in southern Asia revealed that local emissions in western India can affect eastern and southern regions' air quality (Kumar et al., 2015a). As a result, global models, which provide boundary conditions needed by regional air quality models, can significantly affect the simulated results (He et al., 2019).

The Indo-Gangetic Plain (IGP) experiences high levels of air pollution during the post monsoon season (October to early December) due to stagnant meteorological conditions and higher air pollution emissions (Adhikary et al., 2007;Marrapu et al., 2014). Figure 1a shows the averaged Aerosol Optical Depth (AOD) retrieved from Visible Infrared Imaging Radiometer Suite (VIIRS) remote sensing instrument during November 2017 over northern India. The IGP region has the highest AOD values with the largest values in the north-western parts, which is mostly due to crop residue burning (Beig et al., 2020;Jethva et al., 2018;Liu et al., 2018;Venkataraman et al., 2018;Vijayakumar et al., 2016). Kulkarni et al. (2020) found India's north-western agricultural fires could contribute up to 75% of Delhi's $PM_{2.5}$ concentration.

Not only is there significant spatial variation over the IGP, but also $PM_{2.5}$ concentrations change on a daily basis (Fig. 1c). Delhi, the capital of India with annual average $PM_{2.5}$ concentration of 120 $\mu gm^{-3}$ (Amann et al., 2017), experienced a severe extreme air pollution during November 2017. Figure 1c shows the daily averaged $PM_{2.5}$ concentrations measured with the US-EPA instrument located at the US Embassy in Delhi. Daily $PM_{2.5}$ concentrations reached values more than 900 $\mu gm^{-3}$, 15 (37.5) times higher than 24-hour averaged Indian standards (World Health Organization (WHO) guidelines) (WHO, 2006). However, it is clear that no day is compliant with the air quality standard values. After this extreme pollution episode, the Indian government officially initiated a comprehensive air quality plan called the National Clean Air Programme (NCAP) to reduce the air pollution (MoEF&CC, 2019). Different groups have studied this period. Dekker et al. (2019) attributed carbon monoxide (CO) accumulation between Nov. 11[th] and Nov. 14[th] to stagnant meteorological conditions; specifically, low wind speeds and shallow atmospheric boundary layers. Moreover, they argued regional air pollution transport was mostly responsible for this extreme pollution episode (Dekker et al., 2019). However, Beig et al. (2019) concluded biomass burning emissions after post-monsoon crop productions, accompanied with long range transported dust from Middle East led to very high pollution levels although stagnant condition favoured it.

While the current focus of research groups and governments is on PM, ozone concentrations also show high values during the post monsoon season. Figure 1d shows measured daily ozone concentrations at one CPCB station in Delhi; concentrations exceeded India's ozone air quality guidelines. Moreover, the ozone concentrations followed a similar daily variation as PM during November 2017 (Fig. 1d). As a result, extreme pollution episodes cause not only PM-related health issues but also increase the risk of Chronic Obstructive Pulmonary Disease (COPD) (the most important health outcome of ozone pollution) (Conibear et al., 2018b;EPA, 2013).

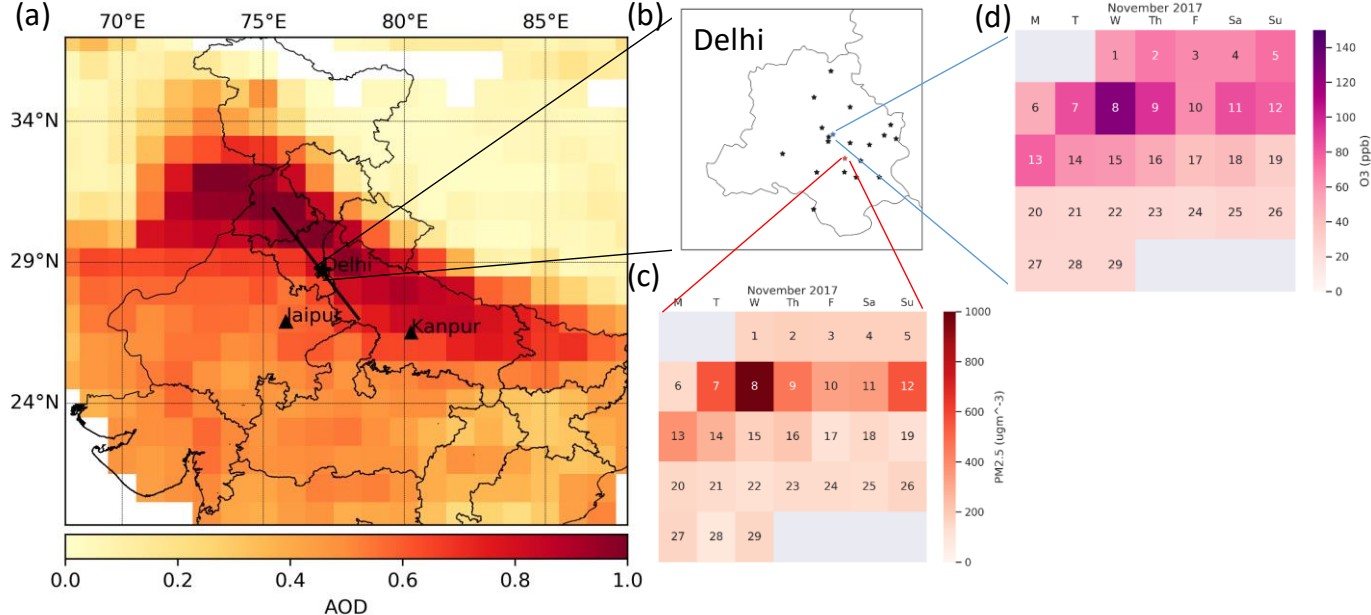

**Figure 1** WRF-Chem modeling domain, ground measurement stations, and observed air quality: a) modeling domain and location of Delhi (∗) and AERONET stations at Jaipur and Kanpur (▲), and underlying VIIRS AOD (550nm) averaged over November 2017, the black line also shows the path that was used for vertical cross section analysis. b) location of CPCB stations (black stars) and US Embassy station (red star), c) Calendar map of averaged daily PM$_{2.5}$ concentration measured at US Embassy, d) Calendar map of averaged daily ozone concentration measured at North Campus, DU

Models usually underestimate the concentrations during extreme pollution periods unless they apply chemical data assimilation (Dekker et al., 2019;Kulkarni et al., 2020;Kumar et al., 2015b;Kumar et al., 2020). Moreover, there are different input data in terms of chemical boundary conditions and fire emissions that can affect air quality modeling results (He et al., 2019). The main purpose of this study is to investigate the sensitivity of model predictions to the main inputs into the model. Prediction of extreme pollution events is important as they have major impacts on people and also make a strong impression regarding the capabilities of models. However, extreme events are hard to predict because they are often heavily impacted by episodic emission sources. Here we take the approach of systematically exploring the impacts of different boundary conditions, dust, fire and anthropogenic emissions on the predictions of the pollution episode in November 2017. A contemporary way to try to capture such events in prediction model is to employ data assimilation (Kumar et al., 2020). The data assimilation results compensate for deficiencies in the inputs as well as structural problems within the models. But the effectiveness of data assimilation improves as the capabilities of the forward model improves. Therefore, our results are also important for those using data assimilation to improve predictability. In this study, we use the Weather Research and Forecasting model coupled to Chemistry (WRF-Chem). Through a series of sensitivity experiments, we evaluate the impacts of biomass burning emissions coming from FINN and QFED, chemical boundary conditions retrieved from MOZART, CAM-chem, CAMS, and MERRA-2 global models, role of incorporating VIIRS active fire hot spots to improve biomass burning emission inventories and global models to improve chemical boundary conditions, and changes in dust and anthropogenic emissions, on modeled PM$_{2.5}$ concentration during November 2017. We also evaluate ozone predictions.

This paper is organized as follows. First, the WRF-Chem model configuration, sensitivity experiments, and the observation datasets, including ground measurements and satellite data, are described. Then, after evaluating the model performance for the best experiment, the impacts of using different datasets as input data on modeled PM$_{2.5}$ concentrations during November 2017 are analyzed and discussed.

## 2. Methods

### 2.1. WRF-Chem configuration:

WRF-Chem is a numerical modeling framework that solves transport, chemistry, and physics of the atmosphere (Grell et al., 2005). The online interaction between meteorology, thermodynamic processes, and atmospheric chemistry makes it a powerful and reliable model in the community. WRF-Chem model (Version 4.0) with 1-domain centered on Delhi with 15 km horizontal grid resolution and 39 vertical levels was used in this study. The domain was set to be big enough to include the north-west biomass burning and urban emission sources in the simulation process as they are shown to be contributors to poor air quality in the region in previous studies (Amann et al., 2017). In the following, we present the model configuration for the base scenario (ID: FINN_VIIRS_7Xperiod2).

National Center for Environmental Prediction (NCEP) Global Forecasting System (GFS-FNL) 1x1-degree and 6-hours spatial and temporal resolution meteorological-fields (https://rda.ucar.edu/datasets/ds083.2/) were used as initial and boundary conditions for the meteorology. Community Atmosphere Model with Chemistry (CAM-chem) data (Buchholz, 2019) with horizontal resolution of 0.9x1.25 degree and 56 vertical levels provided chemical boundary conditions for gaseous species. MERRA-2 reanalysis data with 0.625x0.5 degree horizontal and 72 vertical model levels were used for aerosol species (Bosilovich et al., 2015). However, input data have uncertainties and small uncertainties in nonlinear governing equations of numerical weather predictions can lead to non-negligible errors in results (Xiu, 2010). As a result, re-initialization of NWP models is suggested instead of free runs (Abdi-Oskouei et al., 2020). In this study, the model ran for 30 hours each day starting at 00Z while the first 6 hours data were discarded to account for daily spin-up. Meteorological initial and boundary conditions, and chemical boundary conditions were re-initialized daily at 00Z using global models. However other than for the first cycle in which global models provided initial chemical conditions data, chemical fields from the previous cycle were used as the next cycle's initial chemical conditions. Table 1 summarizes the WRF-Chem physical and chemical configuration options.

Studies have shown improvements for ozone simulations in Delhi using more complicated chemistry mechanisms like Model for Ozone and Related chemical Tracers (MOZART) and CBMZ comparing to simple mechanisms like RACM and RADM (Gupta and Mohan, 2015;Sharma et al., 2017). MOZART gas phase chemistry mechanism and the four-bin Model for Simulating Aerosol Interactions and Chemistry (MOSAIC-4bin) were used for modeling atmospheric chemistry and aerosol properties as suggested in previous studies over India (Kumar et al., 2015a). MOZART, version 4 mechanism was initially developed for global modeling of ozone and other tracers in the troposphere (Emmons et al., 2010). Although it includes 97 gas-phase and bulk aerosol, all monoterpenes, which are important in ozone chemistry, were lumped together. As a result, Hodzic et al. (2015) added a detailed treatment of monoterpenes and Knote et al. (2014) updated the isoprene oxidation scheme in the MOZART mechanism in WRF-Chem. MOSAIC is an aerosol model that considers a wide range of aerosol species that are important in regional scale and treats the chemical and microphysical processes between them including nucleation, coagulation, thermodynamics and phase equilibrium, and gas-particle partitioning (Zaveri et al., 2008). Hodzic and Jimenez (2011) updated secondary organic aerosol (SOA) formation mechanism and the updated version is available in WRF-Chem for doing regional air quality modeling studies. MOSAIC-4bin, used in current study, calculates all the above-mentioned aerosol physics and chemistry in four sectional aerosol size bins with the assumption that each bin is internally mixed and all the particles within a bin have the same chemical composition (Zaveri et al., 2008).

In India, both anthropogenic and natural sources have important impacts on air quality. Biomass and biofuel use in residential sector for heating and cooking purposes have significant contributions to air quality in India (Conibear et al., 2018a;David et al., 2019;Venkataraman et al., 2018). Moreover, there are more than 1000 power plants and brick kilns in India that are major anthropogenic sources for $SO_2$ and particulate matters, respectively (Guttikunda and Calori, 2013). Other than these industrialized

sources, literature shows that biomass burning (e.g. agricultural waste burning) contributes to 37 percent of air pollution over sub-
continent (Kumar et al., 2015a). Hemispheric Transport of Air Pollution (HTAP v2.2) (Janssens-Maenhout et al., 2015) emission inventory of 2010 with 0.1 degree horizontal resolution, mapped to MOZART-MOSAIC mechanism (https://www2.acom.ucar.edu/wrf-chem/wrf-chem-tools-community), was used as the base anthropogenic emission inventory. Although accuracy of urban anthropogenic emission inventories have significant effects on air quality modeling studies (Gupta and Mohan, 2015;Kumar et al., 2012;Sharma et al., 2017), the focus of this paper is to capture the air pollution due to regional
sources; we didn't use higher resolution emission inventories for Delhi.

Fire INventory from NCAR, version 1.5 (FINNv1.5) and Model of Emissions of Gases and Aerosols from Nature (MEGAN v. 2.0.4) were used as biomass burning emission and biogenic emission inventories, respectively (Guenther et al., 2006;Wiedinmyer et al., 2011). However, other studies have noticed that uncertainties in FINN emissions can significantly modify the results (Kulkarni et al., 2020). Therefore, two modifications were applied to FINN data to provide better input data: filling missing fires
using VIIRS Fire Radiation Power (FRP) data and scaling the fire emissions (scaling procedure described in detail later). Liu et al. (2018) used FRP values to approximate the stubble burning areas affecting Delhi's air quality. In their statistical study, 99% of post monsoon FRP values were attributed to agricultural fires (Liu et al., 2018). In this study, we used FRP values to improve fire emissions. Specifically, we first regridded VIIRS 375m resolution FRP data to our domain. Then at each hour, for all grid cells that have FINN emissions, we find the corresponding mean VIIRS FRP, and do a linear regression between FRP and emission
flux. Afterwards, we apply the regression line parameters on VIIRS FRP for the grid cells that do not have any FINN emission, to estimate the flux. It should be mentioned, all the available FRP data were utilized disregarding the retrieval's confidence level. Moreover, we used VIIRS instead of MODIS data as it provided higher resolution active fire points data (375m vs. 1km), which is an important point for small fires. For example, no active fire points in Moderate Resolution Imaging Spectroradiometer (MODIS) instrument were reported in 2018 post-monsoon for Uttar Pradesh (Kulkarni et al., 2020). Figure 9 shows more fire grid
cells in eastern IGP and central India when incorporating VIIRS data to FINN inventory. We acknowledge that this technique is a first-order approximation and can have large errors as FINN is based on burned area algorithms from MODIS retrieved data; more detailed research is required to improve the idea.

Dust storms are an important natural pollution source that have caused many pollution events over some parts of India (Kumar et al., 2014a). Goddard Global Ozone Chemistry Aerosol Radiation and Transport (GOCART) mechanism was used to calculate the
threshold wind velocity and total dust emission, which about 70 percent of total mass was then distributed in different bins of the other inorganics (OIN aerosol component in WRF-Chem; OIN represents all primary inorganic PM) component in the model with the assumption that the rest are larger than $PM_{10}$ (Zhao et al., 2010). This is based on the study in Northern Africa, where Zhao et al. (2010) allocated about 1% in bins with diameter less than 2.5 micron and 69% of the dust in bin 4( 2.5-10 microns), and assumed the rest were bigger than 10 micron and will not remain in the atmosphere for an influential period.

**Table 1** Details of WRF-Chem physical and chemical setup configuration

| Process | Method |
| --- | --- |
| Domain | 1domain (15km horizontal resolution) |
| Landuse | MODIS 20-category |
| TimeStep | 60 seconds based on CFL stability criterion (Courant et al., 1928) |
| Vertical | 39 (top at 5hpa) |
| Microphysics | Morrison double-moment scheme (Morrison et al., 2005) |
| Longwave Radiation | RRTMG, called every 5 minutes |

| Shortwave Radiation | Goddard, called every 5 minutes |
|---|---|
| Planetary Boundary Layer | MYNN-level3 (Nakanishi and Niino, 2009) |
| Land Surface | Noah Land Surface Model (Wang et al., 2018) |
| Gas-Phase Chemistry | MOZART-4, called every 10 minutes |
| Photolysis Scheme | New TUV, called every 10 minutes |
| Aerosol Scheme | MOSAIC 4-bin (no aqueous phase chemistry), called every 10 minutes |
| Dust | GOCART (Ginoux et al., 2001) |
| Initial and Boundary meteorology | NCEP FNL |

## 2.2. Sensitivity experiments

Three sets of experiments were performed to explore the impact of using different global data, as either boundary conditions or emissions, and dust emission formulation on $PM_{2.5}$ and AOD predictions (Table 2). It should be mentioned that all the modeling options and other input data remained unchanged unless specified.

One set of experiments focused on the sensitivity of the predictions to biomass burning emissions. First, we compared the impacts of two different biomass burning emission inventories, namely FINN and Quick Fire Emission Dataset (QFED) (Darmenov and da Silva, 2013). Specifically, simulations using QFED (ID: QFED_CAMCHEM) and FINN (ID: FINN_CAMCHEM) were performed to understand the impact of different fire detection algorithms. When using QFED, it should be mentioned that we mapped total CO values to VOC species in MOZART chemistry mechanism based on emission factors provided in literature instead of using VOCs emissions directly from QFED (Akagi et al., 2011). Second, we investigated whether FINN fire emissions were underestimated for all the days (ID: FINN_10Xall), some days (ID: FINN_10Xperiod1), or just one day before the pollution episode on Nov. 5[th] (ID: FINN_10Xday). Then after modifying FINN using VIIRS FRP data, we did a sensitivity test with changing the period for scaling fire emissions. Specifically, we scaled fire emissions for a 15-days period between Nov. 3[rd] and 17[th] (ID: FINN_VIIRS_10Xperiod1) and an 8-days period between Nov. 5[th] and 13[th] (ID: FINN_VIIRS_10Xperiod2). We also evaluated the performance for a scaling factor of 10 in comparison with 7 (ID: FINN_VIIRS_7Xperiod2). Anthropogenic emissions over India also have high uncertainties (Saikawa et al., 2017). As a result, we studied how increasing the anthropogenic aerosol emissions by a factor of 2 affects the results (ID: BASE_ANTHRO2X).

Another set of experiments evaluated the impacts of chemical boundary conditions. Many global datasets can be used in regional air quality modeling. Simulations were performed using CAM-chem (ID: FINN_CAMCHEM), MOZART (ID: FINN_MOZART), Copernicus Atmosphere Monitoring Service (CAMS, ID: FINN_CAMS.), and a combination of CAM-chem for gaseous and MERRA-2 for aerosol species (ID: FINN_MERRA2) global modeling systems. It is important to note that CAMS and MERRA-2 are reanalysis models and use observed data to improve the results. CAMS assimilates MODIS and Advanced Along-Track Scanning Radiometer (AATSR) satellite instruments AOD (Inness et al., 2019). MERRA-2 assimilates AOD from multiple sources including MODIS, Multiangle Imaging Spectroradiometer (MISR), Advanced Very High Resolution Radiometer (AVHRR), and Aerosol RObatic NETwork (AERONET) although assimilating some products have been stopped after 2014 (Randles et al., 2017). Finally, simulations were conducted for various dust emission modifications. In one simulation, we turned off dust emission option in the model (ID: NO_DUST), while in another simulation, we increased total dust emissions by 5 times to explore if dust emission were underestimated in the model (ID: DUST_5X). Moreover, we changed the allocation of total dust in different bins of MOSAIC module to see whether different allocation of aerosols can contribute to the observed extreme pollution in Delhi (ID: DUST_allocation). Specifically, we reduced 30% from the 4[th] bin (2.5-10 micron) and distributed 25% of it in 3[rd] bin (0.625-2.5 micron) and 5% in 2[nd] bin (0.156-0.625 micron). More allocation to bins 2 and 3 were not considered, as it was not realistic to the

large-size nature of dust aerosols. FINN_10Xall scenario represents the simulation with turned-on dust option (original allocation) in the model. The detailed results from experiments on boundary conditions and dust emissions can be found in the supporting document.

**Table 2** List of scenarios performed in this study

| Simulation ID | Initial/Boundary Chemical (Gaseous / Aerosol) Condition | Biomass Burning Emission Inventory | DUST |
|---|---|---|---|
| **Reference Scenario** | | | |
| **FINN_VIIRS_7Xperiod2 (base scenario)** | CAMchem (gas) + MERRA2 (aerosol) | 7 times higher (FINN+VIIRS) for Nov $5^{th}$ to Nov $13^{th}$ | GOCART |
| **Biomass Burning Emission Sensitivities** | | | |
| **QFED_CAMCHEM** | CAMchem (gas +aerosol) | QFED | GOCART |
| **FINN_CAMCHEM** | CAMchem (gas + aerosol) | FINN | GOCART |
| **FINN_10Xall** | CAMchem (gas) + MERRA2 (aerosol) | 10 times higher FINN | GOCART |
| **FINN_10Xday** | CAMchem (gas) + MERRA2 (aerosol) | 10 times higher FINN for Nov $5^{th}$ | GOCART |
| **FINN_10Xperiod1** | CAMchem (gas) + MERRA2 (aerosol) | 10 times higher FINN for Nov $3^{rd}$ to Nov $17^{th}$ | GOCART |
| **FINN_VIIRS_10Xperiod1** | CAMchem (gas) + MERRA2 (aerosol) | 10 times higher (FINN+VIIRS) for Nov $3^{rd}$ to Nov $17^{th}$ | GOCART |
| **FINN_VIIRS_10Xperiod2** | CAMchem (gas) + MERRA2 (aerosol) | 10 times higher (FINN+VIIRS) for Nov $5^{th}$ to Nov $13^{th}$ | GOCART |
| **Boundary Condition Sensitivities** | | | |
| **FINN_MOZART** | MOZART (gas + aerosol) | FINN | GOCART |
| **FINN_CAMS** | CAMS (gas + aerosol) | FINN | GOCART |
| **FINN_MERRA2** | CAMchem (gas) + MERRA2 (aerosol) | FINN | GOCART |
| **Dust Emission Sensitivities** | | | |
| **NO_DUST** | CAMchem (gas) + MERRA2 (aerosol) | 10 times higher FINN | Turned Off |
| **DUST_5X** | CAMchem (gas) + MERRA2 (aerosol) | 10 times higher FINN | 5 times higher GOCART emission |
| **DUST_allocation** | CAMchem (gas) + MERRA2 (aerosol) | 10 times higher FINN | GOCART put 30% of bin 4 in bins 2 and 3 |
| **Anthropogenic Emission Sensitivity** | | | |
| **BASE_ANTHRO2X** | Similar to Base scenario (ID: FINN_VIIRS_7Xperiod2) except anthropogenic aerosol emissions increased by a factor of 2 | | |

## 2.3. Observation data

The model performance was evaluated using ground measurements, space-borne instruments, and global reanalysis data. Specifically, we used data collected by the Central Pollution Control Board (CPCB) over the domain for doing statistics. It includes stations over Delhi (19 stations), Rajasthan (10 stations), Haryana (4 stations), and Punjab (3 stations). No additional quality control filters, other than the ones by CPCB (https://cpcb.nic.in/quality-assurance-quality-control/), were applied. We evaluated the results after applying the filters proposed by other studies (e.g. Kumar et al. (2020)); they had slight impacts on statistics (shown in the

supporting document). $PM_{2.5}$ data measured by an US EPA instrument at the US embassy in Delhi was used as the reference station. Level-2 VIIRS remote sensing instrument data onboard Suomi-National Polar-Orbiting Partnership (S-NPP) was used for comparing the spatial pattern of AOD and fire counts over the domain. Specifically, aerosol products with around 6km horizontal resolution based on the Deep Blue algorithm (Hsu et al., 2019) and 375 m active fire products based on VNP14IMG algorithm (Schroeder et al., 2014) were used. There are only two AERONET stations in the domain (Fig. 1a). AERONET data at these two sites confirmed the reliability of VIIRS retrieved data (Fig. S5). MERRA-2 gridded data was also used to evaluate the model performance. MERRA-2 reanalysis is based on the assimilation of many meteorological data and the assimilation of AOD from multiple satellites (Gelaro et al., 2017). The on-ground continuous monitoring stations guidelines state that instruments should sample at heights between 3-10m. Irrespective of this condition, some of CPCB stations are placed on top of the buildings with restricted clean flow of air (personal inspections). While we observed little impact of this situation on the concentrations in a well-mixed layer, a meteorological parameter like wind speed data can show erratic behaviour. As a result, we used MERRA-2 meteorological data to evaluate the WRF-Chem simulations using 10m wind speed and direction, 2m temperature, surface water vapor mixing ratio variables.

We also compared MERRA-2 AOD (at 550nm) and $PM_{2.5}$ predictions with WRF-Chem results to evaluate how the assimilation of AOD affected the predictions. The MERRA-2 $PM_{2.5}$ was based on the mass mixing ratios of black carbon, organic carbon, dust, sea-salt, and sulfate. Since ammonium concentration is not available, it is common in the literature to assume that sulfate ion will be completely neutralized by ammonium and form ammonium sulfate and therefore a factor of 1.375 was assumed in calculating inorganic aerosol concentrations (Buchard et al., 2016;He et al., 2019;Provençal et al., 2017). On the other hand, literature suggest organic carbon concentration should be multiplied by 1.4 to compensate for other missing organic compounds to estimate the organic mass (Buchard et al., 2016;Chow et al., 2015;He et al., 2019;Provençal et al., 2017;Turpin and Lim, 2001). However, Turpin and Lim (2001) argued that this scaling factor should be 2.6 for biomass burning particles; we used 2.6 according to our studied time period and potential black carbon sources:

$$[PM] = [BC] + 2.6 * [OC] + 1.375 * [SO_4] + [DUST] + [SS] \tag{1}$$

Where BC is black carbon, OC is organic carbon, $SO_4$ is sulfate, DUST is dust, and SS is sea salt concentrations. As dust and sea salt data are reported in multiple bins, different bins should be used for different particle diameters.

The metrics we used to assess the performance of the simulations are Root Mean Squared Error (RMSE), Mean Error (ME), Normalized Mean Bias (NMB), Normalized Mean Error (NME) and correlation coefficient (R) as defined in the supporting document (Emery et al., 2017;Emery et al., 2001). Since low values can have significant impacts on normalized values, which are used in mean normalized metrics, normalized mean values are better metrics and used in this study (Emery et al., 2017).

## 3. Results and discussions

### 3.1. Model performance

Our analysis between different simulations revealed that FINN_VIIRS_7Xperiod2 scenario had the best statistical performance of the configurations studied. This scenario is called the base scenario and we evaluate it in this section. Performance of the base model in capturing the meteorological parameters was evaluated using MERRA-2 data for 10m wind speed and direction, 2m temperature, and surface water vapor mixing ratio. Figure 2 (a, b, c, d) show these comparisons at the location of the US Embassy in Delhi (28.59° N, 77.19° E). The model was able to capture the general diurnal trend for all these variables and the sharp shift in wind direction between Nov. 13[th] and Nov. 17[th], after the extreme pollution episode. Negatively biased wind speed with ME of 1.1 m/s and RMSE of 1.28 m/s shows the model generally underestimated wind speed and it was most predominant between Nov. 17[th]

and Nov. 25th. Figure 2c shows the model did not accurately capture nighttime 2m temperature minima but captured the maximum values with overall overestimated ME of 3.52° C and RMSE of 4.01° C. The wind speed satisfied the benchmark RMSE value of 2.0 m/s, while temperature was higher than the targeted ME goal of 2.0° C (Emery et al., 2001). The representation error plays an important role in evaluating results due to different horizontal resolutions in the model and MERRA-2 dataset (~0.15x0.15 vs 0.625x0.5 degree), specifically in urban areas. For instance, the same statistics for a rural area in Rajasthan (27.0° N, 73.0° E; not

shown) have smaller biases and are compliant with benchmark values (RMSE of 0.99 m/s for wind speed and ME of 1.08° C for 2 m temperature). For water vapor mixing ratio the model clearly captured the daily variations; specially, the increase after the pollution episode (Nov. 13th). However, it showed a very sharp day-to-night shift during the pollution episode days. The spatial performance of the model averaged over November during daytime hours (8AM to 6PM) is shown in Fig. 2 (panels e, f, g, h, i, j). The sharp gradient between Himalayas and IGP regions in the north-east, the gradient between land and sea in the south-west, and

the slight gradient between different land types in north-west of the domain for both 10m wind speed and 2m temperature were captured well. Overall, the model was able to capture the general daily variations and spatial trends when compared to MERRA-2 data.

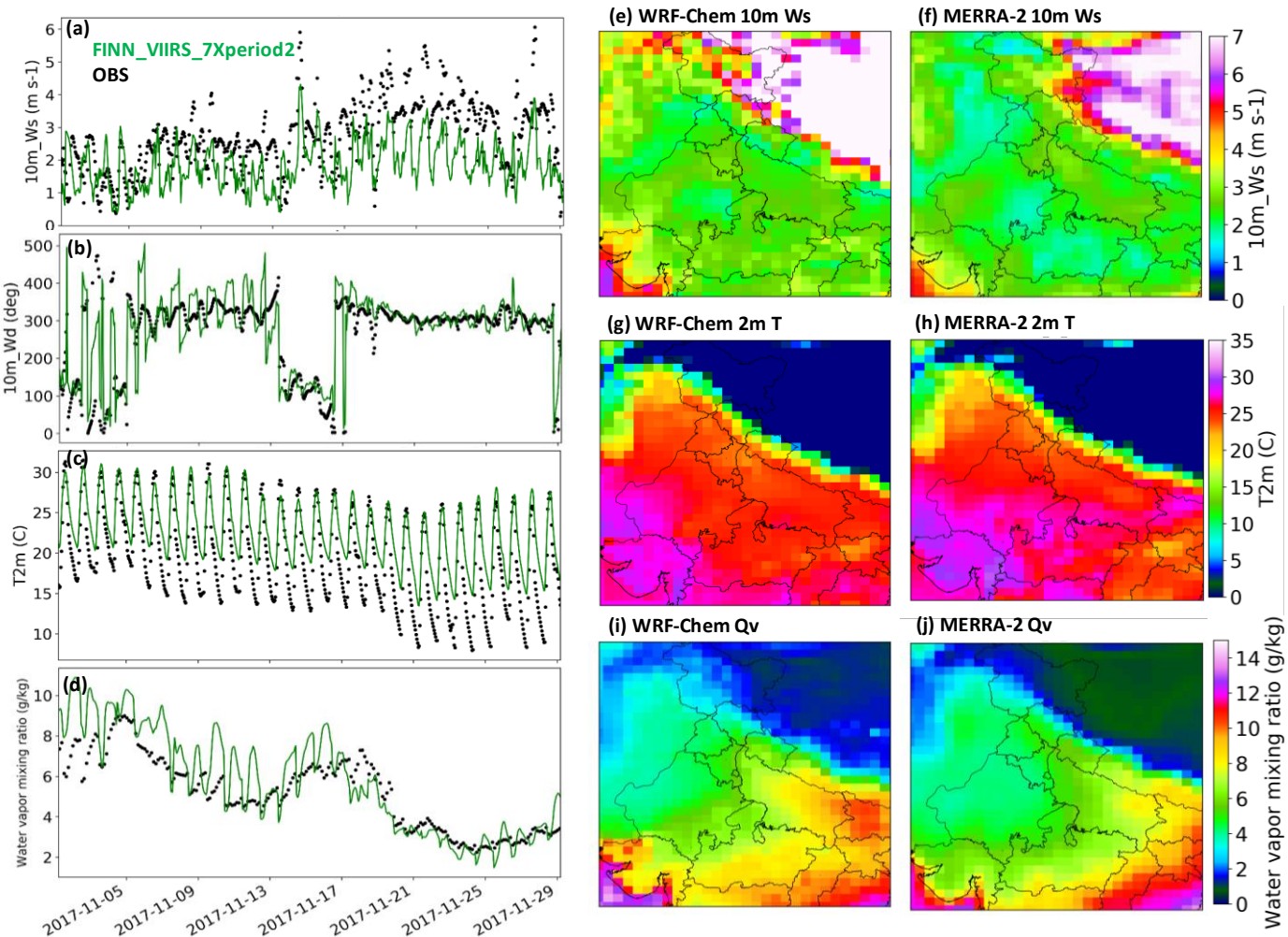

**Figure 2** Temporospatial meteorological performance of base scenario simulation: Time series of simulated (green line) and MERRA-2 (black
dots) hourly a) 10 m wind speed, b) 10 m wind direction, c) 2m temperature, d) surface water vapor at US Embassy coordinates. e,f) Averaged daytime (8AM-6PM) 10 m wind speed maps of modeled (e) and MERRA-2 (f). g,h) Averaged daytime (8AM-6PM) 2 m temperature maps of simulated (g) and MERRA-2 (h). i,j) Averaged daytime (8AM-6PM) surface water vapor mixing ratio (g/kg) maps of model (i) and MERRA-2 (j)

Figure 3 shows spatial distribution of the base scenario, VIIRS data, and the bias for 550 nm AOD, averaged over all the days in Nov., Nov. 5th as a day with intensive fire emissions, and Nov. 24th as an illustrative day after the extreme pollution episode. Model showed high AODs over Delhi and Punjab, confirming satellite data. Moreover, AODs were high over western IGP, close to major fires of Punjab, with a gradual gradient towards eastern and central India. Dust emission sources in the border of Pakistan also led to high AODs although they did not affect Delhi as discussed in the supporting document. In general, the model underestimates AOD over IGP and overestimates elsewhere. WRF-Chem predicted the averaged AOD over the whole domain for Nov. 2017 to be 0.58 ($\pm$0.4), while VIIRS data showed 0.43($\pm$0.26). AOD maps for Nov. 5th show the model generally underestimated AOD values for the entire IGP region, except for Punjab. Moreover, the model underestimated aerosol loadings over central India. Other studies have reported biased low AOD and corresponding $PM_{2.5}$ concentrations over other polluted regions (He et al., 2019;Song et al., 2018). Nov. 24th, on the other hand, represented a day with no significant fire emission. The model did a good job capturing AOD values in the central parts of India and around Delhi. However, the model missed high AOD values in eastern IGP. MERRA-2 data also did not show high AODs over the western border with Pakistan and did not capture extremely high AODs over Punjab, although it showed AOD enhancements (Fig. S6).

Figure 4 shows time series of modeled, MERRA-2 product, VIIRS retrievals, and observed AOD at the AERONET stations (location shown on Fig.1). AOD values at Kanpur, a station in the eastern IGP, were more than 1.0 before the pollution episode and reached up to 2.0 during the episode days, and decreased to values between 0.5 and 1 for the rest of days. The model captured the general trend although missed high AOD's between Nov. 9th and 13th, while MERRA-2 successfully captured the AOD trend through the whole month, including days with enhanced AOD values. This shows that AOD assimilation in MERRA-2 significantly improves AOD predictions. At Jaipur, located in southern IGP, the model overestimated AOD for the first five days of November. During the pollution episode days, the model is biased high compared to MERRA-2 and VIIRS retrievals. AERONET data showed low AOD values before the pollution episode but did not report values during the pollution episode. It suggests, as one possibility, that PM concentrations were too high during this period that the instrument was not able to retrieve data. After the pollution period, AOD values were lower than 0.5, showing relatively low PM concentrations. In general, MERRA-2 showed better performance in terms of NMB (Kanpur: -1.3% and Jaipur:-20.1%) compared with our model (Kanpur:-27.4% and Jaipur: +29.9%). Comparing averaged AOD with VIIRS retrievals for BASE_ANTHRO2X scenario showed lower bias over the IGP (Fig. S7). These results show the need for improved estimates of biomass burning as well as anthropogenic emissions. We also looked at Angstrom Exponent (AE) at Jaipur and Kanpur to understand if the model captured the mode of the particles (Fig. S8). Over Jaipur the model is biased high compared to AERONET data (NMB: 30%) and shows more finer aerosols in the model. After Nov. 20th, both AERONET and VIIRS retrievals suggest the dominance of coarser aerosols, while the AE for the model does not follow the same trend. However, modeled $PM_{2.5}/PM_{10}$ ratio shows more coarse aerosols compared to the rest of the month (Fig. S9). Over Kanpur, the model AE is biased high (NMB: 50.8%). On the other hand, the model shows closer AE values to VIIRS retrievals. For example, both the model and VIIRS retrieval show similar reduction in AE on Nov. 8th and 9th. Kumar et al. (2014b) also reported slight AE overestimation in WRF-Chem during a pre-monsoon dust storm at Kanpur and Jaipur. Furthermore, model and AERONET have variational trend while MERRA-2 is smooth during the whole month at both Jaipur and Kanpur.

Figure 5a shows time series plot of base scenario and observed $PM_{2.5}$ concentration at the US Embassy station. Observed values were high throughout the month on the order of 200 $\mu$gm$^{-3}$ with diurnal variations due to changes in the mixing heights. The extreme pollution episode began on Nov. 7th, when $PM_{2.5}$ concentrations increased to more than 800 $\mu$gm$^{-3}$. On Nov. 8th, the values increased even more to about 1000 $\mu$gm$^{-3}$. $PM_{2.5}$ concentrations started decreasing on Nov. 9th and continued through Nov. 10th. However, values increased again and were high between Nov. 11th and Nov. 13th. Afterwards, they returned to ~ 200 $\mu$gm$^{-3}$ for the

rest of the month. The model accurately captured the magnitude and diurnal cycle for $PM_{2.5}$ for non-episodic days. Moreover, the model was able to see the sharp increase in concentration in the beginning of the episode starting from Nov. 7th with reported $PM_{2.5}$ concentrations of ~650 $\mu gm^{-3}$. This sharp increase was captured after incorporating VIIRS data into FINN emissions accompanied by scaling the emissions by a factor of 7. In fact, increased emissions from fires in agricultural fields in the north-west on previous days and favorable north-westerly winds, as shown on Fig.2 explain this concentration hike. However, the model underestimated the concentrations for the next three days. Then, the model captured the second enhancement. Although wind direction showed good agreement with MERRA-2 dataset and wind speed was biased low and even more favorable for stagnant conditions, modeled $PM_{2.5}$ concentrations had a large negative bias for the period between Nov. 8th and Nov. 10th. This suggests either low local anthropogenic emissions within Delhi or some missing pollution sources upwind of Delhi that were not included in the emission estimates led to underestimation.

Dekker et al. (2019) studied CO concentrations during Nov. 2017 using satellite observations and they reported low emissions as one of the reasons for large negative concentration biases, although they proposed unfavorable meteorological condition as the main reason for high CO concentrations in Delhi, between Nov. 11th and Nov 14th. Moreover, Cusworth et al. (2018) reported that MODIS based biomass burning emission inventories miss many small fires over India. Beig et al. (2019) concluded that long range transported dust coming from Middle East was a major source for this extreme pollution episode. Figure 5a shows that MERRA-2 did not capture high surface $PM_{2.5}$ concentrations after Nov. 8th. Navinya et al. (2020) reported that MERRA-2 underestimates $PM_{2.5}$ over India, especially during post-monsoon season. More discussions on the extreme pollution episode are provided in the following section.

Starting on Nov. 13th, the modeled concentrations went down as winds shifted to easterlies and wind speed increased. Beig et al. (2019) found $PM_{2.5}$ concentrations after the pollution episode were lower compared with similar periods in previous years. Thereafter, the concentrations went back to values for Delhi before the episode. The model did a fairly good job in capturing the trend during non-episode days (Table S4). Increasing anthropogenic emissions (ID: BASE_ANTHRO2X) on simulation results overestimated $PM_{2.5}$ concentrations in the US embassy location during non-episode days (Fig. S7).

Figure 5b and Figure 5c show the averaged $PM_{2.5}$ maps for all the hours over the studied region in the base simulation and MERRA-2 dataset. The model was able to capture higher concentrations over north-western India and the border with Pakistan where agricultural fire and dust emissions play the most important role for extreme pollution episodes over IGP. However, the model showed higher values than MERRA-2 over southern Punjab, the region with high biomass burning emissions (Kulkarni et al., 2020). Since MERRA-2 assimilates satellite AOD data as its major aerosol forcing, it will not be able to capture high concentrations if satellite retrieval algorithms miss corresponding high AODs.

Figure 5b and Figure 5c also show that the model was biased high over central India and biased low over eastern IGP. These results indicate improving emissions in eastern IGP can significantly improve the simulation results. Conibear et al. (2018a) also reported limited success of models to capture the spatial variability of $PM_{2.5}$ over India in 2016, specifically during winter. Table 3 provides statistics for 24-hours averaged $PM_{2.5}$ concentrations for base scenario simulation for Delhi and its western states. Statistics for Delhi show NMB of -16.6%, which passes the "criteria" benchmark of 30% , while NME of 27.6% shows better performance and complies with the benchmark "goal" of 35% for the whole month (Emery et al., 2017). Correlation coefficient of 0.48 is also higher than the benchmark criteria of 0.4. Statistics significantly improve after excluding the four extremely polluted days between Nov. 7th and Nov. 10th and all are within benchmark goals (Table S4). Kumar et al. (2020) assimilated MODIS AOD to WRF-Chem in order to improve the air quality forecasts over Delhi. In their study, Mean Bias for first-day forecast of $PM_{2.5}$ concentration decreased from -98.7 $\mu gm^{-3}$ to -13.7 $\mu gm^{-3}$. They also showed that RMSE decreased from 167.4 $\mu gm^{-3}$ to 117.3 $\mu gm^{-3}$. Our results

from the base scenario (Mean Bias: -42.38 $\mu gm^{-3}$ and RMSE: 118.47 $\mu gm^{-3}$) shows comparable results to the data assimilation technique, while still both models are biased low.

Statistics for Haryana state (4 stations) show good performance (NMB of -7.5% and correlation coefficient of 0.4). The model was biased high for Rajasthan (10 stations NMB: 15.5%) and Punjab (3 stations NMB: 17%). The model slightly overestimated $PM_{2.5}$ concentrations during the episode days in Rajasthan but captured the concentrations during the rest of the month (Fig. S10). In Punjab, measured data did not report $PM_{2.5}$ enhancement during the extreme episode, while the model showed very high concentrations after scaling fire emissions by a factor of 7. However, VIIRS satellite images (e.g. Fig. 9d) clearly show massive

agricultural fires in this state during November and its signals were expected in the measured data. The overall scatter plots including the averaged values for each state shows good spatial performance of the base scenario (Fig. S11).

     Although different meteorological parameters can be responsible for the biases, accuracy of anthropogenic emissions is important. For example, recent local anthropogenic emission inventories developed for Delhi have higher particle emissions than in the regional inventory used in this study, which impacts modeled $PM_{2.5}$ concentrations for typical days (Kulkarni et al., 2020). We

conducted BASE_ANTHRO2X scenario to investigate the effect of uncertainties in the anthropogenic emissions. This scenario increased $PM_{2.5}$ concentrations in Delhi up to ~150 $\mu gm^{-3}$, which led to overestimation (in contrast to underestimation in base scenario) at many of non-episode days (Fig. S7). Although this scenario did not help in capturing the high concentrations during the episode, it confirms the need for better anthropogenic emissions. On the other hand, it reduced the bias over IGP (Fig. S7). These results point out the need for best estimates of emissions of both anthropogenic and biomass. .Maps also show that averaged

$PM_{2.5}$ concentrations over most of India were higher than the air quality standard.

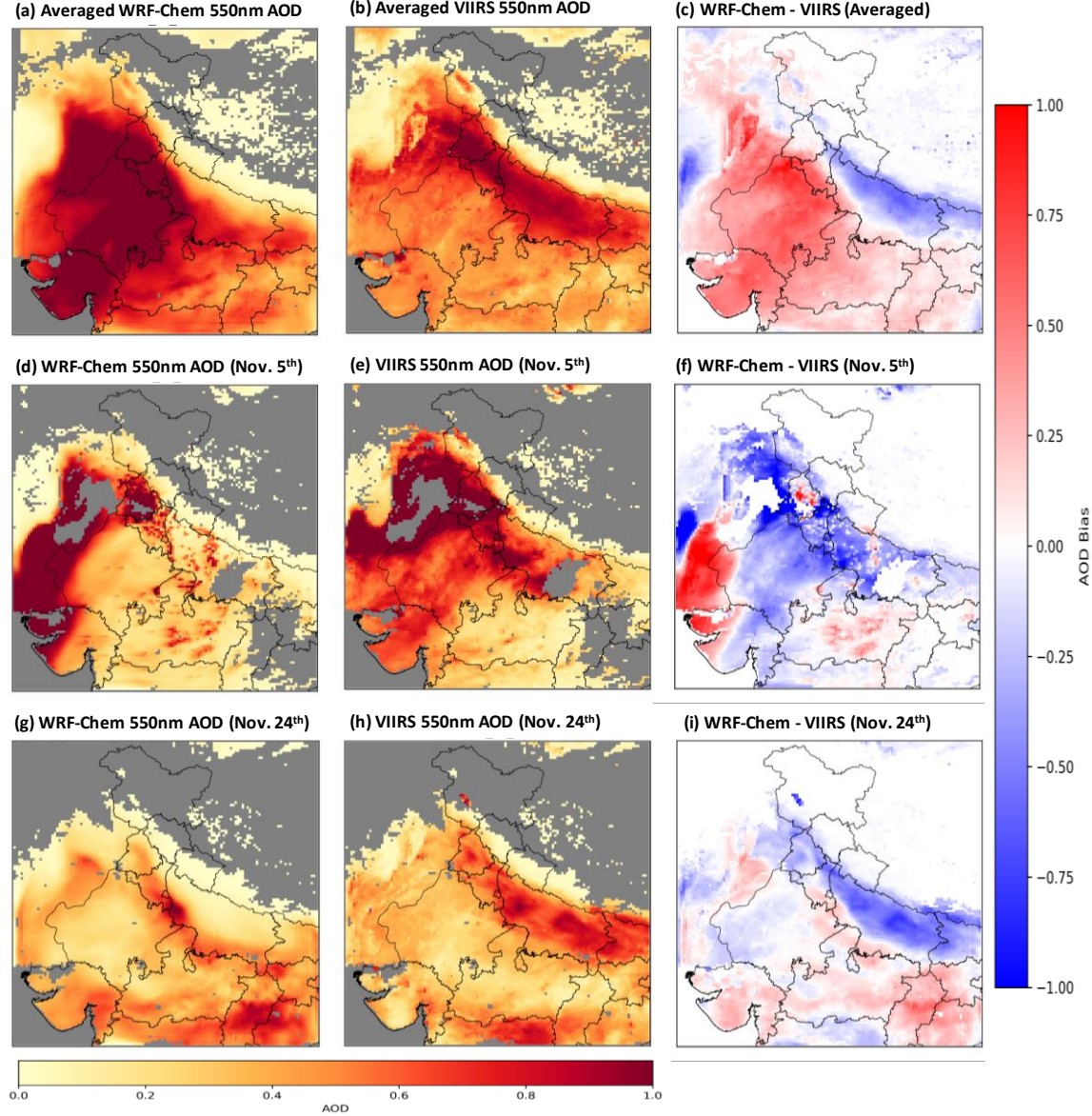

**Figure 3** Spatial distributions of AOD at 550 nm averaged over whole November (top panel), Nov. 5th (middle panel), and Nov. 24th (bottom panel). WRF-Chem maps represent base scenario results. Differences between model and VIIRS are also shown.

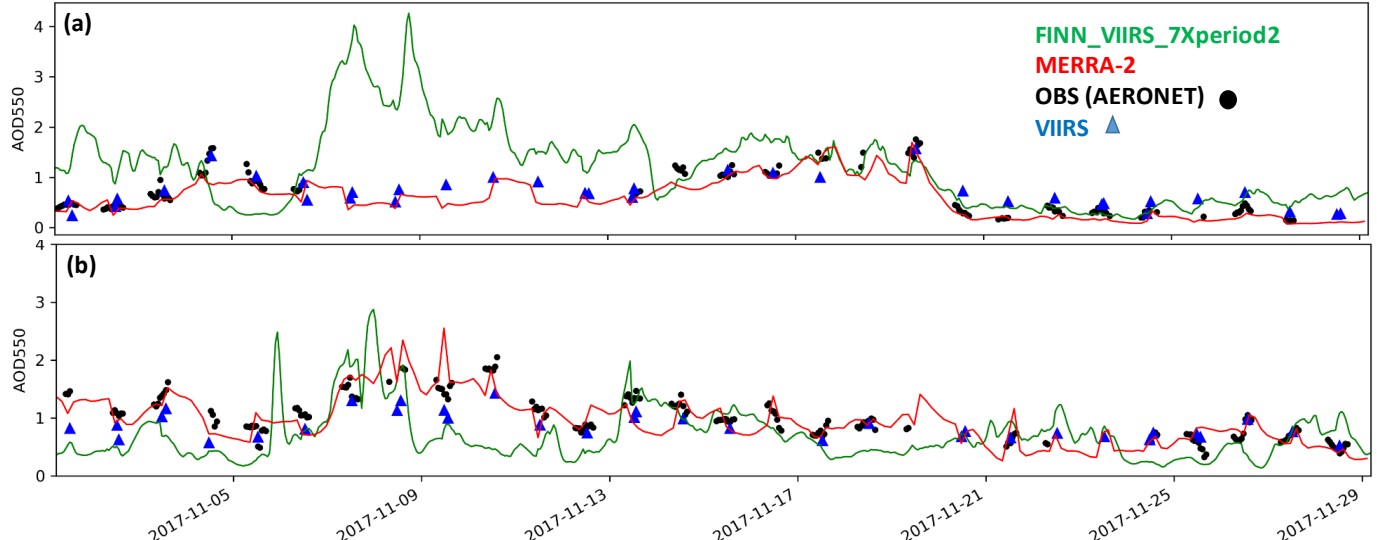

Figure 4 Time series of modeled (green line), VIIRS retrievals (blue triangle), MERRA-2 (red line), and AERONET (black dots) AOD at 550 nm during Nov. 2017 at a) Jaipur, b) Kanpur.

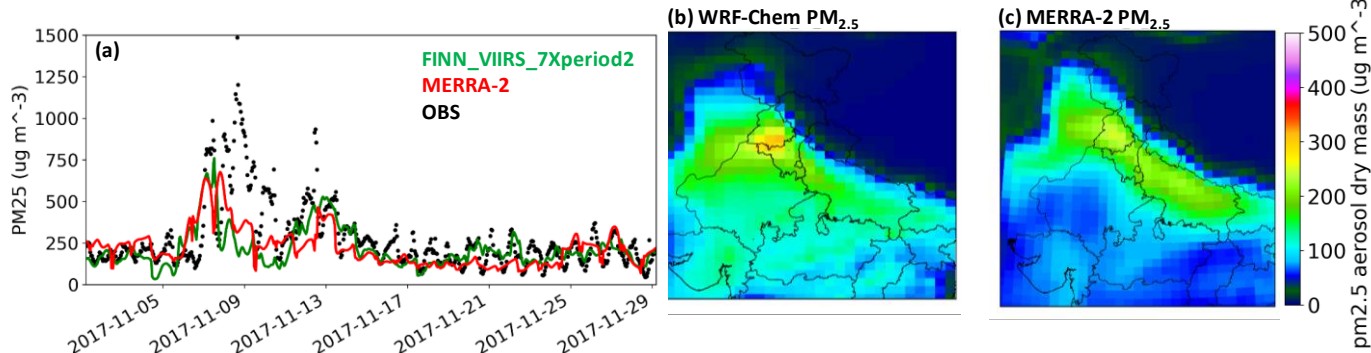

Figure 5 Temporospatial air quality performance of base scenario simulation: a) Time series of simulated (green line), MERRA-2 (red line), and ground measurement (black dots) hourly PM$_{2.5}$ concentration at US Embassy coordinates. b, c) Hourly averaged PM$_{2.5}$ concentration maps of model regridded to MERRA-2 resolution (b) and MERRA-2 (c).

Table 3 Mean (±standard deviation), Normalized Mean Bias (NMB), Normalized Mean Error (NME), and Pearson Correlation Coefficient (R) averaged for all CPCB stations in different states during Nov. 2017. Model values are for base scenario (FINN_VIIRS_7Xpeiod2). Mean values are for hourly data, while NMB, NME, and R relates 24-hours averaged values.

| State | Hourly Obs. Mean (±std) ($\mu gm^{-3}$) | Hourly Model Mean (±std) ($\mu gm^{-3}$) | 24-hours NMB (%) | 24-hours NME (%) | 24-hours R (%) |
|---|---|---|---|---|---|
| Delhi | 255.5 (±146.6) | 213.9 (±113.9) | -16.6 | 27.6 | 0.48 |
| Haryana | 177.7 (±77.6) | 165.8 (±89.9) | -7.5 | 29.5 | 0.40 |
| Punjab | 139.9 (±54.7) | 166.7 (±198.3) | 17 | 55.5 | 0.24 |
| Rajasthan | 123.4 (±62.7) | 147.7 (±62.7) | 15.5 | 34.4 | 0.22 |

## 3.2. Extreme pollution episode analysis

Figure 6a shows the box and whisker plots for daily PM$_{2.5}$ concentration for the base scenario and all of the CPCB stations in Delhi. 24-hour averaged measured values over all CPCB stations in Delhi for PM$_{2.5}$ ranged between 133 $\mu gm^{-3}$ and 664 $\mu gm^{-3}$, which is

about 2 and 11 times, respectively, higher than India 24-hours standard value of 60 $\mu gm^{-3}$. The model showed overall good performance for daily PM$_{2.5}$ concentrations for non-extremely polluted days (Table S4: NMB of -2.44 and R of 0.7) and followed the observed trend in the extreme pollution episode (Fig. 6), which suggests the overall meteorology and transport patterns were captured by the simulations. However, the model started the episode on Nov. 6th and significantly overestimated the concentrations. The model captured the median for Nov. 7th very well, although measured values span a wider range. The model missed the high concentrations on Nov. 8th, which led to underestimations on Nov. 9th and Nov. 10th, as well, regardless of capturing the decreasing trend. However, the model was able to simulate the second wave of the episode starting on Nov. 11th and accurately captured the median and range of PM$_{2.5}$ concentrations on Nov. 12th and Nov. 13th. It is important to point out that the underestimation of PM on the 9th and 10th persisted for all of the sensitivity cases performed. This suggests either the transport in the model during these days missed high source regions and/or significant emission sources for these days were not included in the inventories.

Back trajectories can be used to provide insights into modelled concentrations during the extreme pollution episode. Back trajectories were calculated for releasing 10,000 air parcels at 100 m above ground level and over eastern Delhi using the FLEXible PARTicle dispersion model (FLEXPART) with inputs from WRF-Chem model output (Brioude et al., 2013). Figure 7 shows 72-hours mean back trajectory maps for Nov. 6th, 7th, 8th, 9th, and 10th. The releasing times are 00 (red line) and 12 (blue line) UTC on each day. Also plotted are the fire (grey line) and anthropogenic (black line) emissions along the trajectory. The model started to build up PM$_{2.5}$ concentrations on Nov. 6th and was biased high (Fig. 6a). Back trajectories for Nov. 6th_00 show PM$_{2.5}$ concentrations were majorly due to anthropogenic emissions (Fig. 7a). However, Nov. 6th_12 trajectories in Fig. 7c show a spike in fire emissions on the previous hours (backward hours: 5 and 30), which immediately led to high PM$_{2.5}$ concentrations. Moreover, trajectory paths for this day reveal that emissions belonged to fires east of Delhi. Figure 9 shows that the fires east of Delhi in the base scenario are due to incorporating VIIRS data into the fire emissions. Therefore, high biased PM$_{2.5}$ concentration may be related to the scaling factor applied to eastern Delhi fires. On Nov. 7th, the model perfectly captured PM$_{2.5}$ median (Fig. 6a). Back trajectories for Nov. 7th_00 (Fig. 7d,e) show the beginning of a shift in wind direction and PM$_{2.5}$ concentration was exclusively due to fire emissions on Nov. 5th (backward hour: 40). Compared to Nov. 7th_00, fire emission footprints for Nov. 7th_12 trajectories are smaller, while higher for local anthropogenic emissions (Fig. 7f). Back trajectories for Nov. 8th show the northern parts' contribution for both releasing times, although trajectories for Nov. 8th_00 crossed through central parts of Punjab. Moreover, local anthropogenic emission sources affected Nov. 8th_00 trajectories. The model underestimated PM$_{2.5}$ concentrations on Nov. 8th, which can be partly related to errors in transport as the trajectories for Nov. 8th_12 crossed eastern parts of Punjab. However, other physical processes or lower anthropogenic emissions can also be responsible for low bias. Delhi's air quality in Nov. 9th_00 was still getting affected by northern parts, while trajectories for Nov. 9th_12 shifted towards the east. Since, trajectories for Nov. 9th do not show any fire or anthropogenic emissions' pulse, either the model missed the dynamics of that day or emission sources. Nov. 10th trajectories show eastern flow, again, and no fire emission contribution.

To further understand the regional scale transport of the smoke plumes, we plotted cross section of PM$_{2.5}$ over the path from Punjab through Delhi (Fig. 8, path line shown in Fig. 1). PM$_{2.5}$ concentrations showed typical values on Nov. 5th_00 although they still exceeded the standard limits. On Nov. 5th_12, concentrations significantly increased over Punjab area because of fires and the winds brought them on a path towards Delhi. The Punjab's smoke did not completely cross Delhi yet on Nov. 6th as back trajectories for 00 and 12 UTC hours also showed the effects of anthropogenic emissions and fires in eastern Delhi. On the other hand, a significant amount of smoke was above the boundary layer as shown in Nov. 6th_12 panel. Due to shifting winds on Nov. 7th (as shown in Fig. 2), the smoke upwind of Delhi blew over Delhi and led to extremely high concentrations. Although the model captured the median in Nov. 7th, it missed the maximum extent of observed values. Cross sections on Nov. 8th, 9th, and 10th show the residual Punjab's smoke in the boundary layer, while we saw the model underestimated PM$_{2.5}$ concentrations on these days.

Measured PM$_{2.5}$ concentrations over Delhi show a decreasing trend between Nov. 8$^{th}$ and Nov. 10$^{th}$ (Fig. 6). Vertical profiles for the base scenario also show the model captured high biomass burning emission period on Nov. 6$^{th}$ (Fig. 12). However, it also showed high amounts of smoke above the PBL. Cross sections for Nov. 11$^{th}$ to Nov 14$^{th}$ can be found in the supporting document (Fig. S12). These results suggest that plume rise in the model released the emissions too high or the model did not mix the smoke down fast enough. Vijayakumar et al. (2016) showed agricultural fires can transport via upper troposphere and subside over Delhi using ECMWF map. Social reasons can also be important as the first reaction of people during hazy days is to drive to work which directly (exhaust emission) and indirectly (road dusts) worsen air pollution.

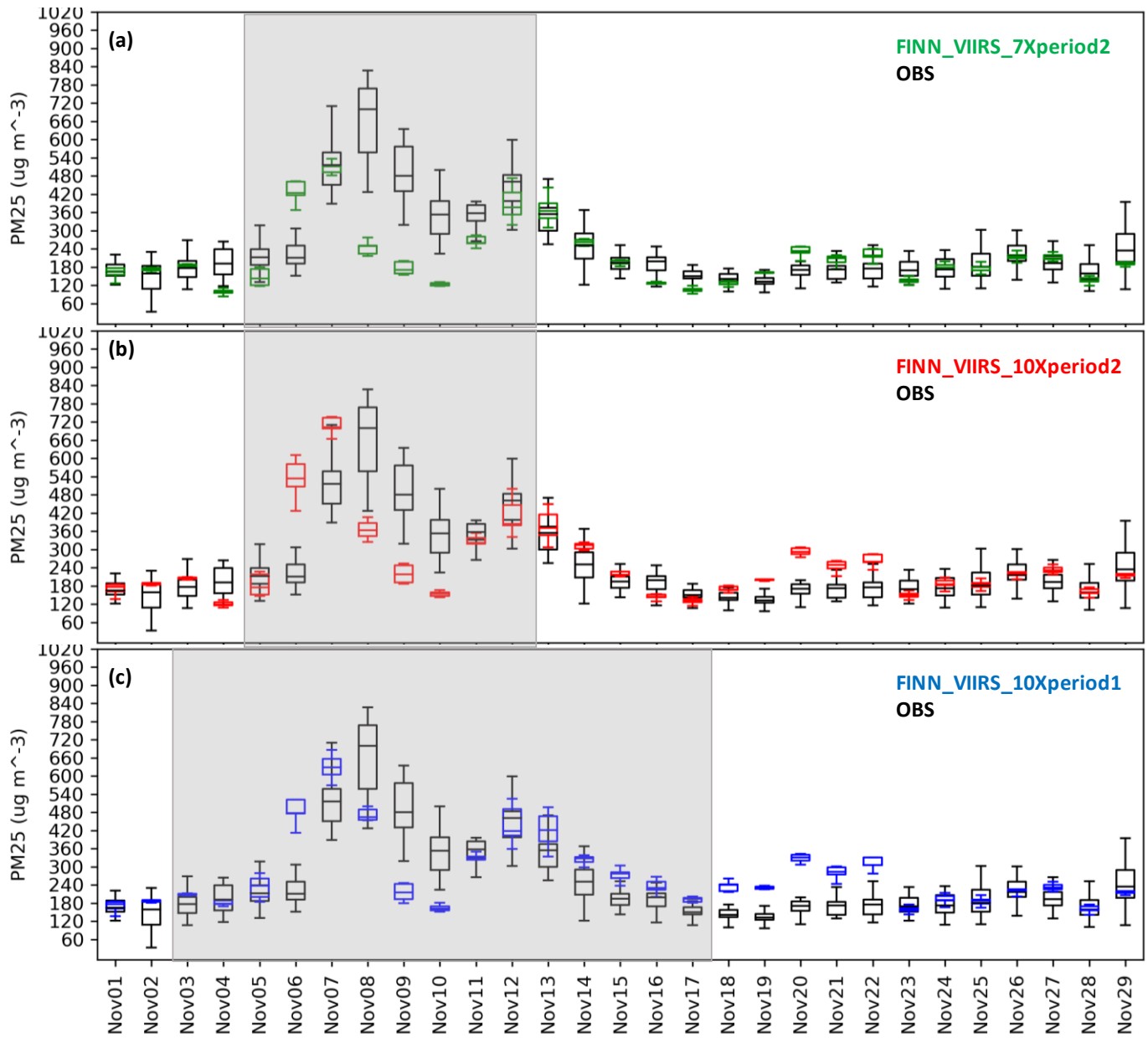

**Figure 6** Box and Whisker plots of observed (black) and modeled daily PM$_{2.5}$ concentration averaged over all CPCB stations in Delhi: a) FINN_VIIRS_7Xperiod2, b) FINN_VIIRS_10Xperiod2, c) FINN_VIIRS_10Xperiod1. Shaded area show the time window that biomass burning emissions were increased.

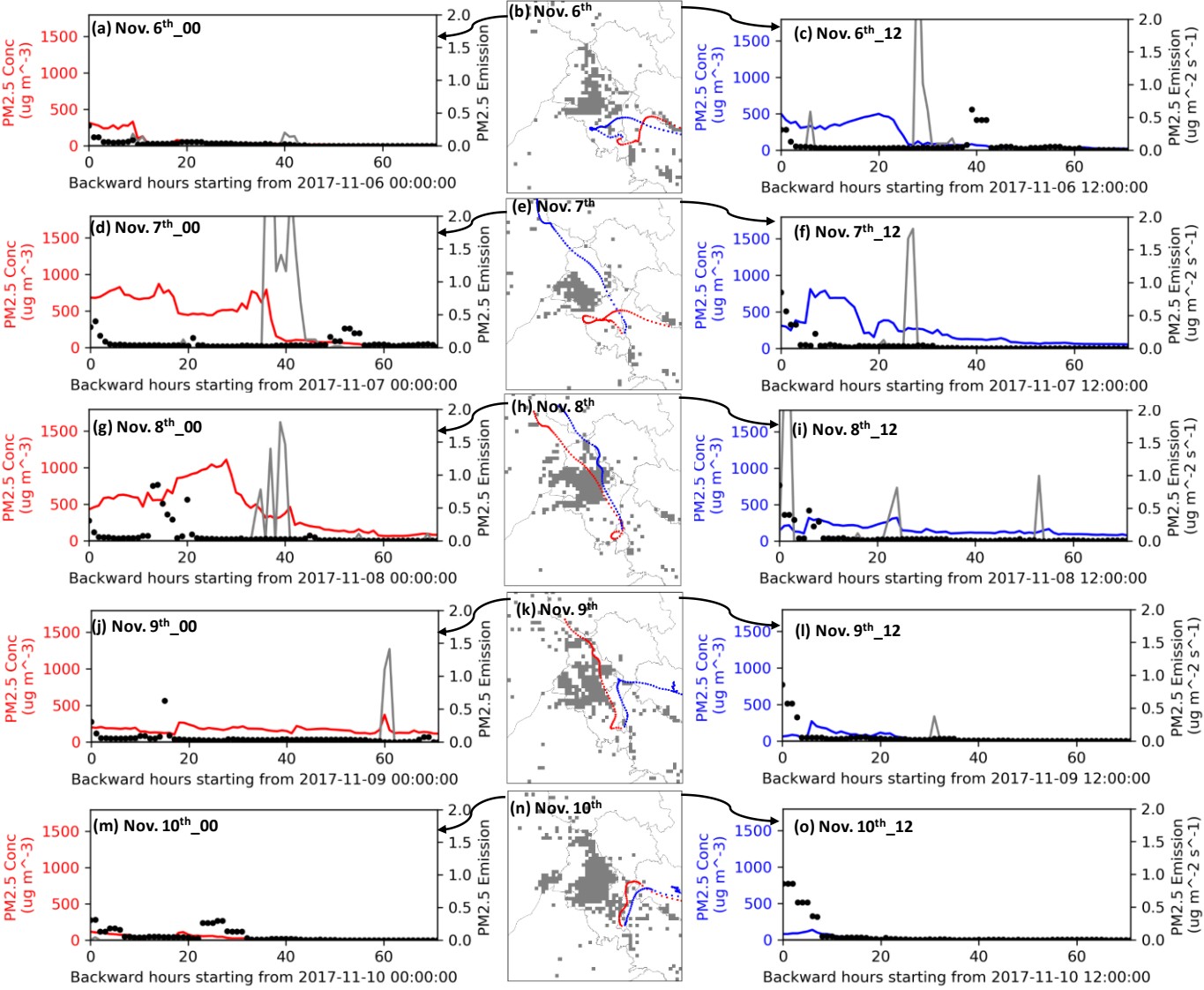

Figure 7 Back trajectory plots of PM$_{2.5}$ concentration (base scenario) on different days for 72 hours: In each row, the map shows back trajectory path for the mass mean location for releasing 10000 particles on eastern Delhi at 00 UTC (red line) and 12 UTC (blue line), where the underlying map shows FINN grid cells on that day. Time series show PM$_{2.5}$ concentrations (primary y-axis) and emissions (secondary y-axis) for Anthropogenic (black dots) and FINN (gray line) inventories along the path. a, b, c) November 6$^{th}$. d, e, f) November 7$^{th}$. g, h, i) November 8$^{th}$. j, k, l) November 9$^{th}$. m, n, o) November 10$^{th}$.00 and 12 UTC denote 5:30 AM and 5:30 PM local time, respectively.

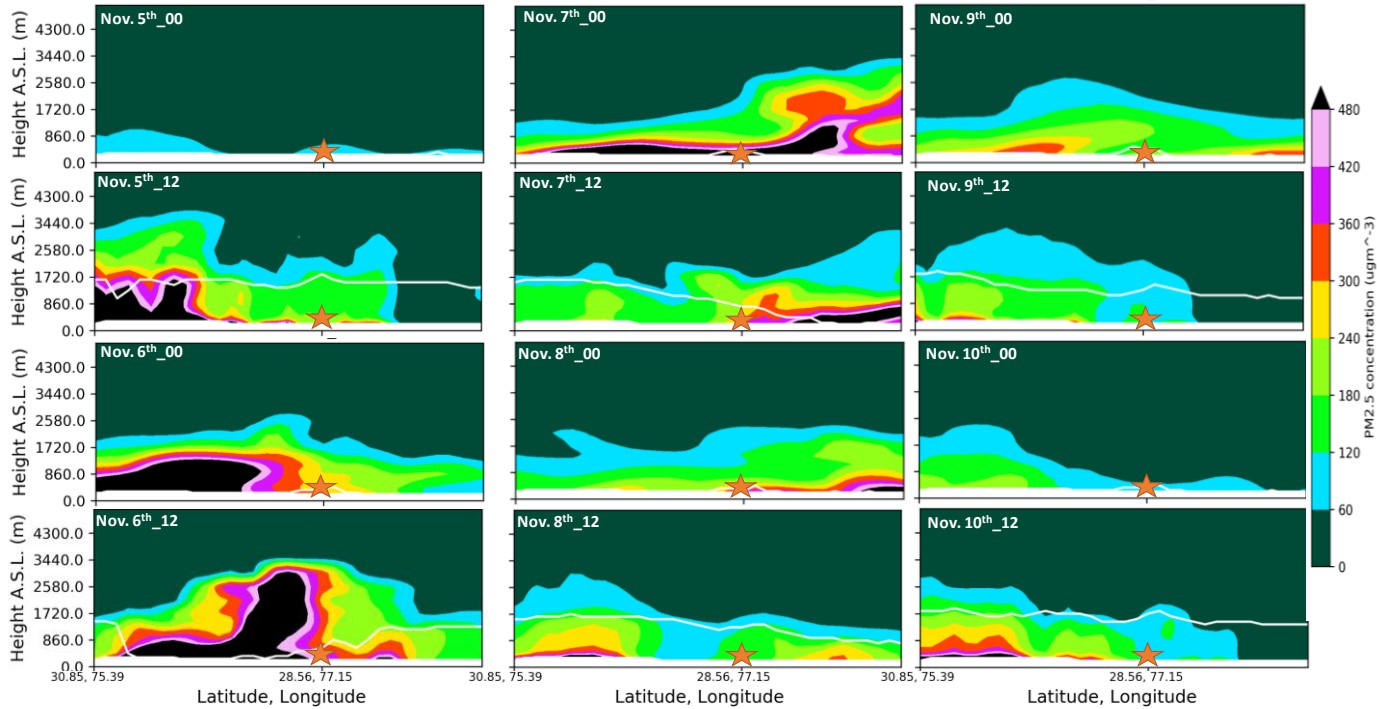


**Figure 8** Vertical cross section of PM$_{2.5}$ concentration through the path shown in Fig. 1 for the days between Nov. 5$^{th}$ and Nov. 10$^{th}$. For each day, two snapshots are shown at 00UTC (5:30AM local time) and 12UTC (5:30PM local time). The orange star shows the location of Delhi through the path. White line shows the PBL height across the path.

### 3.3. Sensitivity to changes in biomass burning emission inventories

Biomass burning emissions used in the base scenario in order to capture the extreme pollution episode were tuned after exploring how these inventories influenced PM$_{2.5}$ concentrations (Table S3). First, we looked at two different emission inventories based on different methodologies and horizontal resolutions; Specifically, FINNv1.5 and QFEDv2.5r1. Both inventories rely on MODIS data; FINN is based on active fire points and estimates of burned area, whereas QFED uses FRP approach (Pan et al., 2020). Figure 9 shows the grid cells with biomass burning emissions based on QFED (panel a), FINN (panel b), and FINN_VIIRS composition

(panel c), used in base scenario, accompanied with active fire points seen by VIIRS (panel d) based on the Fire Information for Resource Management System (FIRMS) product for Nov. 5$^{th}$. It shows FINN captured more fire points in the domain although missing some in eastern IGP and central India while QFED missed almost all of the fire points in Punjab on that specific day. As a result, the QFED simulation did not show any major signal for PM$_{2.5}$ concentration on Nov. 7$^{th}$, whereas the experiment using FINN inventory (ID: FINN_CAMCHEM) followed the measured start of the episode period, regardless of its low bias (Fig. S13).

In general, results using QFED inventory had worse statistics (Fig. 11 and Table S3), which is mostly due to the inability of the inventory to capture the fire points over the domain and it can be attributed to both the technique and the resolution as QFED data have ~10 km resolution, whereas FINN data has ~1 km resolution. Pan et al. (2020) found high uncertainty between different biomass burning emission inventories over Southeast Asia. They showed FINN is, in general, a better dataset for tropical regions as its 2-days continuous fire emission compensates for the lack of daily MODIS coverage used in QFED (Pan et al.,

2020;Wiedinmyer et al., 2011). Dekker et al. (2019) increased the GFAS biomass burning emission inventory by 5 times and did not see any improvement on CO simulation and reported about 2 percent contribution from fires on Delhi's air quality. Our results confirms that FINN provides better biomass burning emissions for India for this period and sheds light on the importance of choosing proper biomass burning emission inventory for a specific domain.

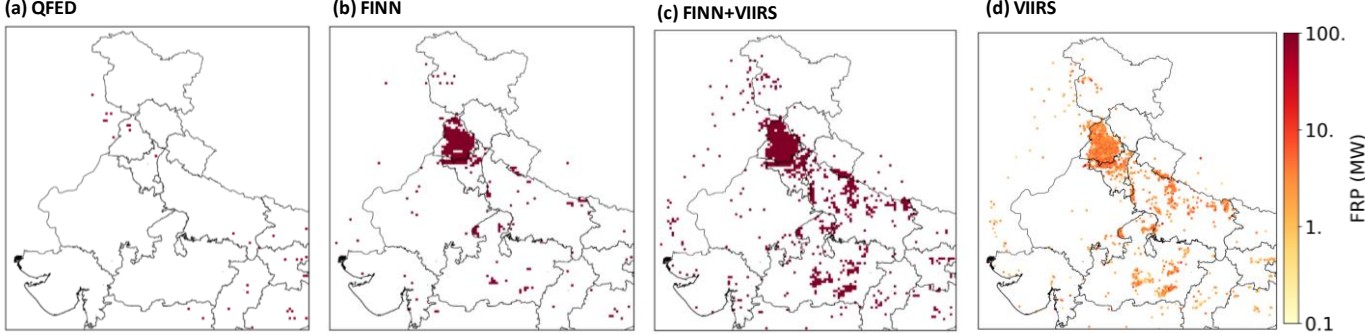

**Figure 9** Spatial fire coverage in different datasets for Nov. 5th: a) QFED fire grid cells b) FINN fire grid cells c) FINN fire grid cells filled missing points with VIIRS d) VIIRS active fire points and corresponding FRP values.

However, the signals from the simulation using FINN biomass burning emission inventory were not high enough as it recorded a maximum concentration of 400 $\mu gm^{-3}$ while the corresponding measured value was 680 $\mu gm^{-3}$. Since observation data are sporadic over India and there were not many ground measurement stations available, sophisticated techniques such as inversion modeling

were not feasible (Saide et al., 2015). As a result, manual tuning of the emission data was performed. The first attempt was to understand if FINN required to be increased for the whole month, a 15-days period around the episode, or just on Nov. 5$^{th}$, which had many fire points in Punjab (Fig. 9). Figure 10 shows PM$_{2.5}$ time series averaged over all CPCB stations based on these scenarios. Increasing FINN emissions for the whole month (ID: FINN_10Xall) led to an overestimation in the first 5 days of November but it significantly helped capturing high peaks on Nov. 7$^{th}$ and 8$^{th}$. Moreover, it increased the concentrations on Nov 12$^{th}$ and 13$^{th}$

regardless of missing the peaks. However, it did not show any improvements between Nov. 9$^{th}$ and 12$^{th}$, which suggests that the included fires did not influence Delhi's air quality during this period. On the other hand, increasing FINN emissions data by 10 times for all days led to very high PM$_{2.5}$ concentrations on later days (Nov 20$^{th}$-27$^{th}$). It showed that FINN data were not systematically biased low. In other words, these results suggest that FINN algorithm underestimated the magnitude of only some fires emission amounts. Some studies have shown that thick fires can be identified as clouds in retrieval algorithms (Dekker et al.,

2019;Huijnen et al., 2016). As another experiment, we increased FINN emission only on Nov 5$^{th}$ since that day had original high values in the inventory (ID: FINN_10Xday). This experiment resulted in better PM$_{2.5}$ concentrations on the last third of November. However, it captured only high concentrations of Nov. 7$^{th}$ and missed the peak of Nov. 8$^{th}$ as well as underestimated on some other days including Nov 13$^{th}$. Finally, we multiplied the fire emissions by 10 for a 15 days period between Nov. 3$^{rd}$ and 17$^{th}$ (ID: FINN_10Xperiod1). In this way, we were able to capture the peaks on Nov. 7$^{th}$ and 8$^{th}$, see major contributions between Nov. 12$^{th}$

and 14$^{th}$, and realistic values between Nov 19$^{th}$ and 27$^{th}$. It should be mentioned this 15-days period was chosen arbitrary and better scaling factors could be achieved by implementing tracers in the model, which is beyond the scope of this paper.

Although this experiment significantly improved the statistical performance for the CPCB stations, the model was spatially underestimating concentrations over eastern IGP (Fig. S14). Moreover, we observed lower fire grid cells in FINN inventory comparing to VIIRS active fire points (Fig. 9). As a result, we tested how incorporating VIIRS data into FINN in order to fill

missing fire grid cells would improve the results. Figure 10 shows how incorporating VIIRS data improved the performance on Nov. 12$^{th}$ and Nov. 13$^{th}$ (ID: FINN_VIIRS_10Xperiod1). However, the model started the episode too early on Nov. 6$^{th}$ and overestimated PM$_{2.5}$ concentrations after the episode. This suggests that the 15-days period for increasing FINN emissions could be too long; we then changed the scaling period from 15 to 8 days between Nov. 5$^{th}$ and Nov. 13$^{th}$ (ID: FINN_VIIRS_10Xperiod2). This modification led to higher PM$_{2.5}$ concentrations on Nov. 7$^{th}$ as Fig. 6 shows. Moreover, the model was still biased high on

after-episode days, which led to choosing the increasing factor of 7 (ID:FINN_VIIRS_7Xperiod2) as our best experiment.

Figure 11a shows the Taylor diagram for hourly PM$_{2.5}$ concentrations based on the studied experiments, representing their statistical performance, for all the days in November. It shows that switching between different experiments mostly improved the standard deviation values. The QFED_CAMCHEM experiment had the lowest standard deviation, but missed high PM$_{2.5}$ concentration values. On the other hand, three experiments using VIIRS-integrated fire emissions had closer standard deviations to measured value. Although the base scenario had good statistical metrics, standard deviation and correlation coefficient were lower, compared to other two VIIRS-included and BASE_ANTHRO2X scenarios for all the days. The reason is that overestimation of other scenarios for Nov. 7$^{th}$ and after-episode days compensate for underestimation between Nov. 8$^{th}$ and Nov. 10$^{th}$. Figure 11b shows the same variables for all the days except Nov. 8$^{th}$, 9$^{th}$, and 10$^{th}$. It shows that the base scenario had the best statistical performance for non-extremely polluted days.

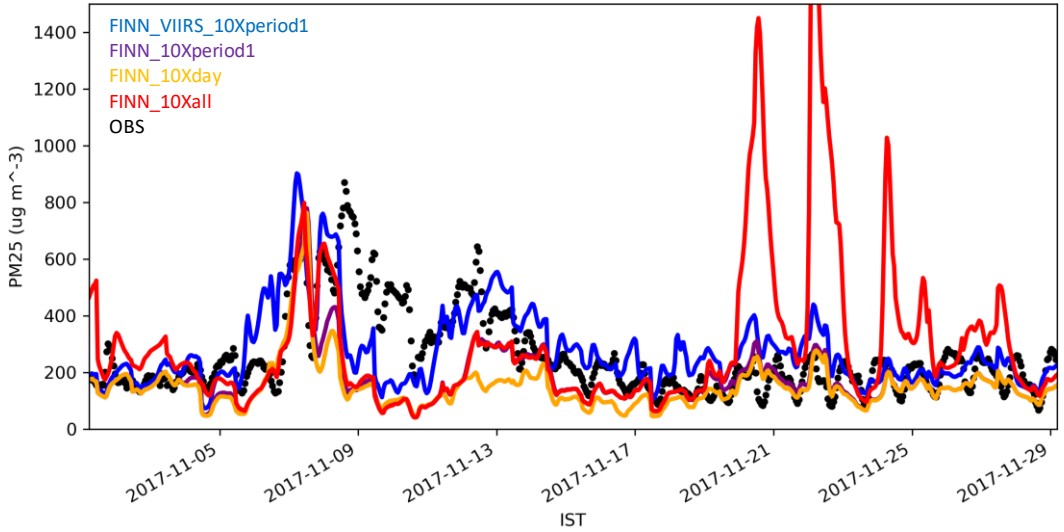

**Figure 10** Time series of model PM$_{2.5}$ sensitivity to 10 times increase in FINN emission inventory for different periods: FINN_VIIRS_10Xperiod1: Nov. 3$^{rd}$ to 17$^{th}$, FINN_10Xperiod1: Nov. 3$^{rd}$ to 17$^{th}$, FINN_10xday: Nov 5$^{th}$, FINN_10Xall: all days. Black dots presents ground measurements data averaged over all CPCB stations in Delhi.

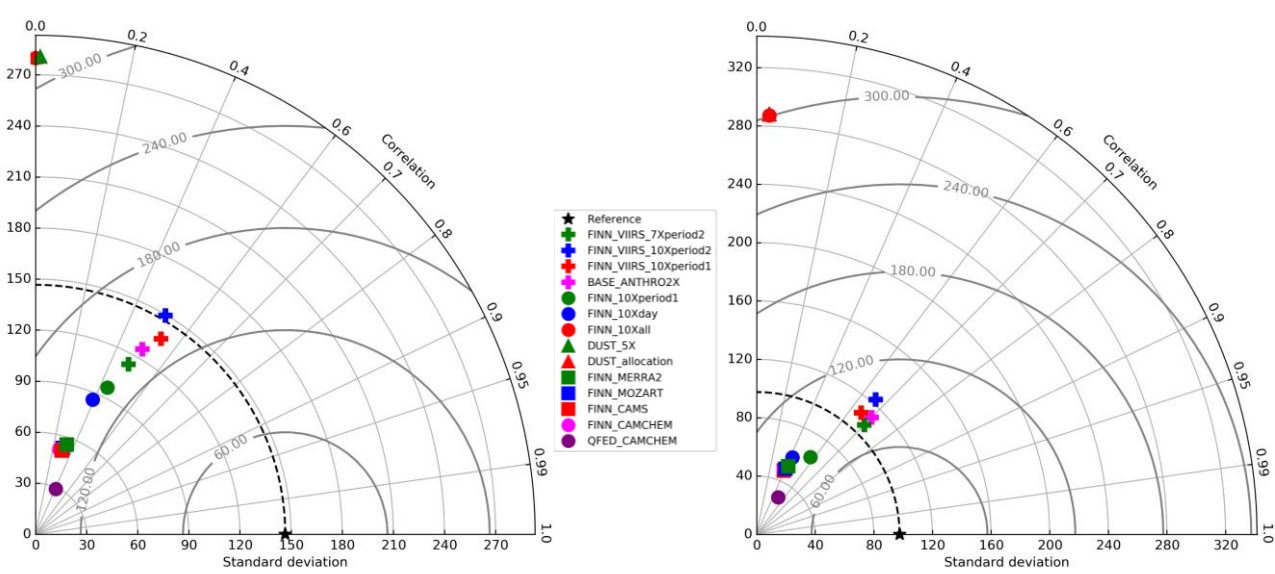

**Figure 11** Taylor Diagram of hourly PM$_{2.5}$ concentration based on different simulation scenarios for: a) all the days, b) all the days except Nov. 8$^{th}$, 9$^{th}$, 10$^{th}$. Plus signs denote experiments after incorporating VIIRS data to FINN, Circles denote different experiments on biomass burning emission inventory, triangles denote dust emission experiments, and squares denote chemical boundary condition experiments. Green colors

show best performance in each experiment. Black star denotes standard deviation of all CPCB stations averaged in Delhi. The statistics plotted are standard deviation on the radial axis and Pearson correlation coefficient (r) on the angular axis, and the gray lines indicate normalized Centered
RMSE (CRMSE).

Figure 12 shows vertical profiles of $PM_{2.5}$ as a function of time at the US Embassy coordinates for variations using the FINN inventory and MERRA-2. Increasing the emissions in FINN inventory significantly increased $PM_{2.5}$ concentrations both vertically and temporally. Although the concentrations got closer to MERRA-2 data, the timing for the peak of boundary layer on Nov. 6[th] was different. By incorporating VIIRS data to FINN and adding more fire emissions, the boundary layer values peaked on Nov.
6[th] earlier and looks much more like MERRA-2 data. Figure 12 also shows more particles at altitudes above the boundary layer, which do not influence surface concentrations. Furthermore, it suggests the scaling factor for the base scenario could be smaller than 7 if the aerosols had been in the boundary layer. It can be partly related to the plume rise module in the WRF-Chem model that may have emitted species at too high altitudes. Increasing emissions also indirectly influenced modeled air quality over Delhi. As our model configuration included feedbacks, absorbing aerosols in the atmosphere (products of fire emissions) decreased the
surface solar radiation budget, changed the dynamics of the atmosphere, reduced the Planetary Boundary Layer (PBL) height, and increased aerosol concentrations. In other words, higher PBLH leads to lower concentrations. For example, Murthy et al. (2020) found that $PM_{2.5}$ concentration decreased up to 14 $\mu g m^{-3}$ for 100m increase in PBLH. Figure 12 also shows the interactions between PBLH and $PM_{2.5}$ concentration at the location of the US embassy. By increasing FINN inventory by 7 times, the PBL height decreased by ~50% on Nov. 6[th], (compare FINN_VIIRS_7Xperiod2 and FINN_MERRA-2 panels in Fig. 12). However, a
measured PBLH dataset can provide better insights. As a result, another study is required to compare modeled PBL heights to observed data (e.g. Nakoudi et al. (2019)) and study the effects of different PBL parameterization modules on aerosol concentrations. Vertical profiles for other experiments using FINN can be found in the supporting document (Fig. S15).
We also did two sets of experiments to understand if long-range transported dust from middle east or in-boundary dust emissions impacted air quality in Delhi. Our sensitivity tests suggest that dusts had a very low contribution to air quality in Delhi during Nov.
2017. Detailed discussion on their impacts have been presented in the supporting document.

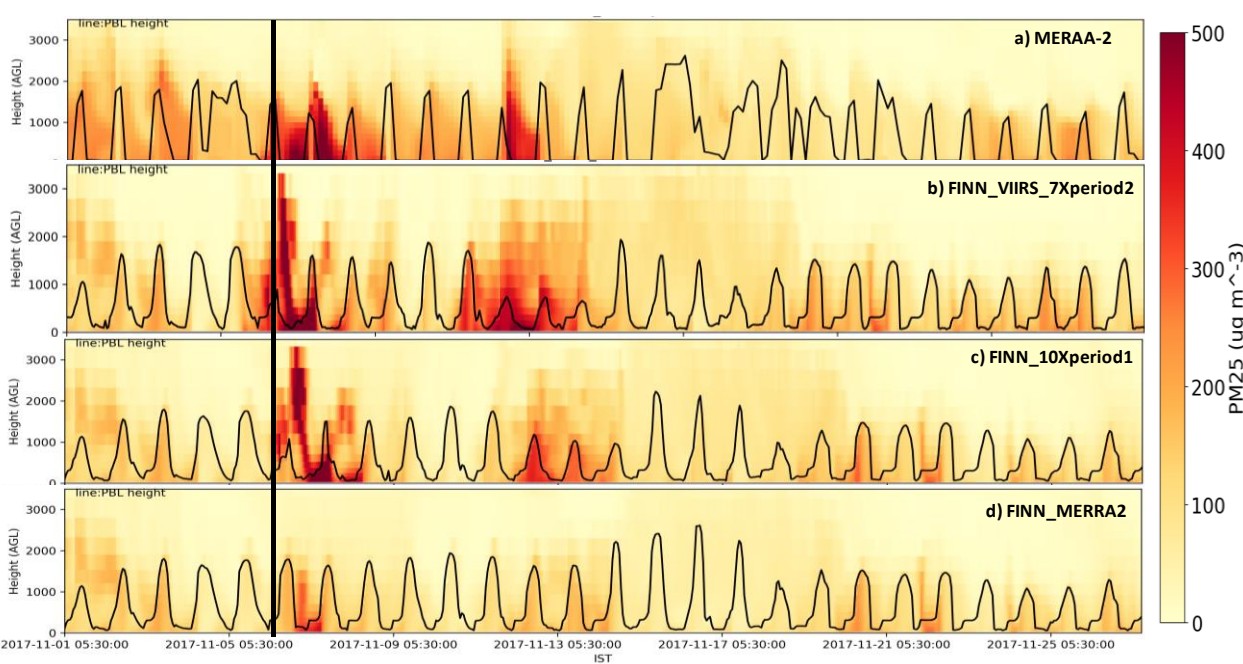

**Figure 12** Vertical cross section of $PM_{2.5}$ sensitivity to major changes in FINN emission inventory at US Embassy coordinates: a) MERRA-2 data as true values, b) FINN_VIIRS_7Xperiod2, c) FINN_10Xperiod, d) FINN_MERRA2. Black lines present planetary boundary layer height. Vertical black line crossing all panels shows boundary layer peak on Nov. 6[th] for MERRA-2 and other experiments.

## 3.4. Aerosol composition in Delhi

We analyzed the modeled $PM_{2.5}$ composition, both in concentrations and by mass fraction, at the location of the US Embassy in Delhi (Fig. S16). Secondary aerosols (secondary organic aerosols (SOAs) + secondary inorganic aerosol (SIA) consisting of Ammonium ($NH_4$) + Nitrate ($NO_3$) + Sulfate ($SO_4$)) comprised 57% of the total averaged $PM_{2.5}$ concentration, whereas primary aerosols (BC + Organic Carbon (OC) + OIN) constituted the rest. Gani et al. (2019) measured $PM_1$ in Delhi and reported 50-70% for secondary aerosols, and $PM_1$ constituted ~85% of $PM_{2.5}$ concentration. SOAs, individually, comprise 27% of the aerosol mass, while SIAs account for 30% of the mass. Amongst inorganic species, NO3, NH4, and SO4 comprise 19%, 7%, and 4%, respectively. Gani et al. (2019) reported the same ranked order but with different percentages. Major contribution of $NO_3$ in winter is also reported in other studies (Pant et al., 2015). BC fraction was 7%, which is very close to the measured fraction of 6.4% in wintertime $PM_1$ (Gani et al., 2019). Pant et al. (2015) reported averaged OC and elemental carbon concentrations of 104.4 $\mu gm^{-3}$ and 46.3 $\mu gm^{-3}$, respectively, which is consistent with our OC/BC ratio of 2.72. Comparing modeled BC1 data with available data for this period (Gani et al., 2019) shows an overall measured to modeled ratio of 1.22, which is consistent with the range other studies reported (Kumar et al., 2015b;Moorthy et al., 2013).

## 3.5. Ozone concentration analysis

Figure 13a shows the box and whisker plots for daytime (8AM-6PM) ozone concentration for the base scenario and all of the CPCB stations in Delhi. Observed values ranged between 10 ppb and 110 ppb and the range between lower and upper quartiles was about 20 ppb, showing high ozone variability over Delhi. Moreover, observed values were higher during the extreme pollution episode. It indicates particles are not the only issue during PM pollution episodes in Delhi. The modeled median was in the range of observed values, especially on non-episode days. However, the model overestimated ozone concentrations on Nov. 7th. Moreover, the range of observed ozone concentrations were wider than modeled values. In general, model captured the trend fairly good with the correlation coefficient of 0.57, but was biased high with NMB of 18% for daytime hours throughout whole November. High biased ozone concentration in Delhi is reported in other studies (Gupta and Mohan, 2015).

Figure 13b shows the daytime ozone concentration maps averaged over November 2017. Central regions of India show higher ozone concentrations compared to northern IGP region. On the other hand, ozone concentration in urban regions were lower than rural areas. This is due to lower ozone production in higher NOx emission regions in urban areas (Ghude et al., 2016;Karambelas et al., 2018). Averaged ozone concentration over the domain throughout Nov. 2017 was 77 ppb using the base scenario. Ozone concentrations decreased by up to 27 ppb when using a scenario without any modifications to aerosol emissions (Fig. S17). Regardless, the averaged values are 9-17 ppb higher than annual averaged concentration of 60 ppb in year 2011 (Ghude et al., 2016). Overall, high measured and modeled ozone concentrations and positive correlation with $PM_{2.5}$ are concerning and demand for more studies. Moreover, recent observed values over Delhi indicated that, during the COVID-19 pandemic that all activities were suspended, $PM_{2.5}$ concentration went down while the trend and range of ozone concentration remained unchanged (Jain and Sharma, 2020).

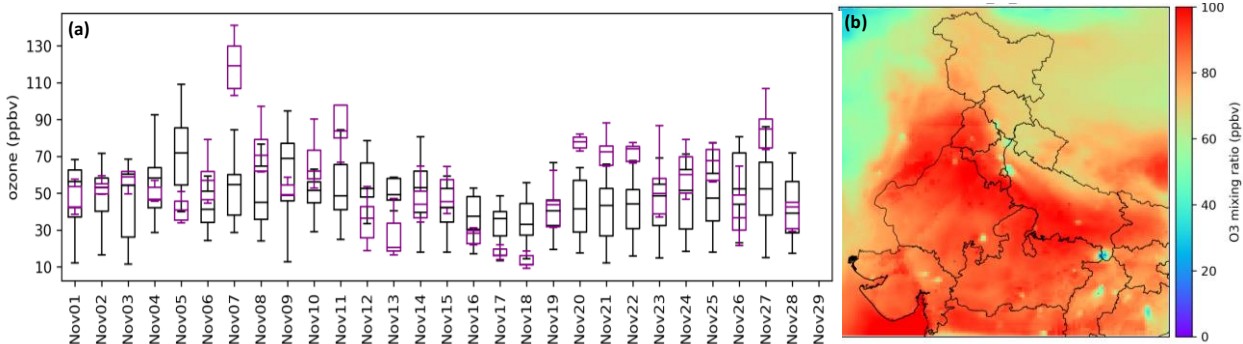

**Figure 13** Daytime (8AM-6PM) ozone modeling performance in base scenario: a) Box and Whisker plots of observed (black) and modeled (purple) concentrations averaged over all CPCB stations in Delhi. b) November 2017 daytime averaged concentrations.

### 3.6. Study limitations

In this study, we used a simple framework to modify Fire emissions with satellite data. Specifically, we used VIIRS data to fill FINN emissions, which are based on MODIS retrievals. We used VIIRS data as they had higher resolution (375m) for active fire points. Furthermore, we used linear regression to find the relation between VIIRS FRP and FINN emissions of available grid cells and applied that to FRP values in VIIRS to estimate the emissions. We acknowledge these are first estimates and the performance of this technique using MODIS data and more complicated statistical works need to be studied further.

During this study, we did not primarily focus on improving anthropogenic emissions over the region in order to capture extreme pollution episode. However anthropogenic emissions are low in global emission inventories and needed to be improved (Jat et al., 2020). Moreover, very low biased concentrations for some days and trajectory results suggest the existence of some other sources, primarily anthropogenic sources, upwind of Delhi that should be studied more.

Furthermore, geostationary satellites can significantly improve our technique as more retrievals could improve the accuracy. In this study, VIIRS or (MODIS) provided only one or two retrievals in one day for each point, while recently launched geostationary satellites, such as GEMS, would provide high temporal frequency data that could improve emission inventories.

The choice of the scaling factor for increasing fire emissions was arbitrary in this study. Due to scarcity of observation data, we were not able to apply complicated mathematical scaling techniques based on data assimilation to scale the fire emissions (Saide et al., 2015). Low number of observation data also limited our statistical assessments. Agricultural fire emissions are small and vary day to day and atmospheric dynamics can significantly change their fate. We did not focus on physics and dynamics of the WRF-Chem model as they were beyond the scope of this study. These are important limitations that readers have to keep in mind when exposed to the results.

### 4. Summary and conclusion

In this study, we used WRF-Chem model to improve the air quality modeling during extreme pollution episode in November 2017 in the IGP. Various modifications on chemical boundary conditions and biomass burning emissions were tested. Multiple datasets, including ground measurements of $PM_{2.5}$, surface measurements and satellite AOD, and reanalysis models were used to evaluate the model. In our best scenario, CAM-Chem and MERRA-2 global models provided gaseous and aerosol chemical boundary conditions, respectively. Moreover, active fire points in VIIRS remote sensing instrument were used to fill the missing fire emission sources in FINN biomass burning emissions. Furthermore, the modified FINN emissions were scaled by a factor of 7 for an eight-days period to capture peak $PM_{2.5}$ concentrations. 24-hours averaged NMB, NME, and R averaged for all CPCB stations in Delhi during all the days in November were -16.6%, 27.6%, and 0.48, respectively, satisfying suggested benchmark criteria (Emery et al., 2017). These metrics significantly improved when excluding four extremely polluted days between Nov. 7[th] and Nov. 10[th] and

all were within benchmark goals (Emery et al., 2017). Overall, we improved modeling results by incorporating different available datasets with each other.

The spatial performance of the model was also evaluated using VIIRS AOD. The model overestimated AOD over the domain with monthly averaged value of 0.58 (±0.4), confirming other studies (Kulkarni et al., 2020). Specifically, the model captured high AODs over Delhi and Punjab, overestimated AODs over central India, and underestimated AODs over eastern IGP. Our results indicate improving emissions, mostly anthropogenic emissions, in eastern IGP can significantly improve the air quality predictions.

Our modeling results revealed secondary aerosols comprised 57% of total $PM_{2.5}$ concentration during November, confirming measurement studies (Gani et al., 2019). Secondary organic aerosols individually, comprised 27% of the total aerosol mass, while secondary inorganics accounted for 30% of the mass.

Back trajectories and vertical profiles were used to study the extreme pollution episode sources. Back trajectories showed a shift in trajectories from east to north on Nov. 7[th]. As a result, agricultural fire emissions were transported from Punjab to Delhi. The trajectories remained on north path for 3 days and then shifted again to east. However, the model underestimated the concentrations on these days. Vertical profiles showed a lot of smoke above boundary layers. These results indicated either the plume rise in the model released the emissions too high, or the model did not mix the smoke down fast enough. Social reasons can also add to high $PM_{2.5}$ concentrations during extreme pollution episodes, as people prefer to use their personal vehicles more often.

We also evaluated how QFED and FINN biomass burning emission inventories affected $PM_{2.5}$ concentration results over Delhi. QFED had worse statistics, which is mostly due to the inability of the inventory to capture the fire points over the domain. It can be attributed to both the technique and the resolution as QFED data have ~10 km resolution, whereas FINN data has ~1 km resolution, as other studies have shown FINN provides better data for India (Pan et al., 2020). We also found FINN underestimated fire emissions for some extremely high emission days, and needed to be scaled. It can be mostly because satellite retrievals reported thick smokes as clouds and missed them, as shown in other studies (Dekker et al., 2019).

The base scenario was chosen after evaluating the results for various chemical boundary conditions, including CAM-Chem, MERRA-2, MOZART, and CAMS global models. We found long-range transported dust from middle-east was not affecting Delhi's air quality during the extreme pollution episode. Moreover, we found MERRA-2 provided better aerosol products over India, although studies have shown they underestimate over India (Navinya et al., 2020). We also found in-domain dust emission sources in the border with Pakistan did not affect Delhi's air quality.

While the focus of current study was on PM, we found high ozone concentration in northern India. Averaged daytime ozone concentration over the domain was 77 ppb for November 2017, using the base scenario. Although the model overestimated ozone concentrations in Delhi by NMB of 18%, it indicates ozone is a problem that needs to be considered.

In general, air quality in IGP region is influenced by both local and regional sources. Although availability of new satellites such as GEMS, which covers some parts of India, can improve air quality predictions using data assimilation techniques, local emission inventories can vary day-by-day and significantly affect the modeling results. More works are required to quantify these impacts. Moreover, ozone concentrations showed a positive correlation with $PM_{2.5}$ over IGP. It suggests that control strategies should consider the regional co-benefits of $PM_{2.5}$/ozone perturbations simultaneously, which is the focus of our future work.

*Data Availability* The WRF-Chem output results for aerosol species are available from Iowa Research Online at https://doi.org/10.25820/data.006126 (Roozitalab et al., 2020).

*Author Contributions* BR and GRC designed the study; BR performed model simulations and analyzed the data with help from GRC and SKG provided measurements. BR and GRC wrote the paper with inputs from SKG.

*Competing Interests* The authors declare that they have no conflict of interest.

*Acknowledgements* We acknowledge the use of data and imagery from LANCE FIRMS operated by NASA's Earth Science Data and Information System (ESDIS) with funding provided by NASA Headquarters. Moreover, this paper contains modified Copernicus Atmosphere Monitoring Service Information 2017, where neither the European Commission nor ECMWF is responsible for any use that may be made of the information it contains. This work was funded in part by NASA HAQAST and ACMAP projects under awards NNX16AQ19G and 80NSSC19K094, respectively. The authors also thank Swagata Payra, Brent_Holben, S._N._Tripathi, and their staff for establishing and maintaining the Jaipur and Kanpur AERONET stations data used in this investigation. We thank Yannick Copin for his python code for plotting Taylor Diagrams. We also thank Dr. Pablo Saide for his comments on using QFED emission inventory.

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
