# Peer review of "Improving regional air quality predictions in the Indo-Gangetic Plain - Case study of an intensive pollution episode in November 2017."

_Atmospheric Chemistry and Physics, 2020_

## Referee Comment (RC1) · Anonymous Referee #1 · 5 Oct 2020

Review of "Improving regional air quality predictions in the Indo-Gangetic Plain – Case study of an intensive pollution episode in November 2017" by Roozitalab et al. (Paper #acp-2020-744).

In this study, the authors have used the WRF-Chem to simulate the intensive pollution episode in the Indo-Gangetic Plain (IGP) in November 2017. They carried out 14 sensitivity simulations for different scenarios based on biomass burning emissions, chemical boundary conditions, and dust emissions. The model (base scenario) was evaluated for meteorological parameters (10 m wind speed and direction, 2 m temperature, and surface water vapor) with MERRA-2. The simulated AOD and $PM_{2.5}$ were compared with observations from AERONET and CPCB/US Embassy monitors, respectively. The authors have also looked into the daytime variation in ozone. The study is interesting because the authors try to simulate the $PM_{2.5}$ during November 6-13 using the emissions and aerosol-radiation feedbacks but no assimilation. This study is similar to a recent study published in JGR by Kumar et al., 2020 (https://doi.org/ 10.1029/2020JD033019). Overall, the manuscript needs a major revision. There are too many figures, which makes it hard to get the message across the reader. The labels in the figures are difficult to read. Here are my comments:

Main comments:

1.  I do not understand the hypothesis behind 14 simulations and still not being able to simulate the aerosols. The authors consider FINN_VIIRS_7Xperiod2 (base scenario) as the best scenario but the bias is still high (for AOD and $PM_{2.5}$) compared to the observations. From Fig. 3, the simulated AOD is underestimated over the IGP and overestimated over the rest of India. Comparison with AERONET (Fig. 4) shows that MERRA-2 does better over Jaipur because from Fig. 3b it is evident that AOD over Jaipur is in the range 0.5-0.8. Both WRF-Chem and MERRA-2 should have the same resolution while comparing (Fig. S2) and color bar scale comparable to Fig. 3. In conclusion, I think AOD is better simulated by MERRA-2 at both Jaipur and Kanpur. Please include the statistics for AERONET vs WRF-Chem and MERRA-2 AOD.
Lines 254-255: There are no major fires during November over western India/Rajasthan (as seen in Fig. 10). Also, there is no major dust event but there is a possibility of anthropogenic dust some of it being unique to the Indian region.
Lines 270-271: I do not agree with the authors' explanation. As seen in Fig. 3, WRF-chem is simulating higher AOD values over western India/Rajasthan.
The figures do not completely agree with the statements made by the authors.

2.  Why are the authors comparing the diurnal variation (Fig. 5a)? Do the emissions have a diurnal variation in the model? Why wasn't the $PM_{2.5}$ data from CPCB stations used in Fig. 5a? Fig. 6 includes data from all the CPCB stations, how was the quality check performed on the CPCB data? Please add the details on the quality check of CPCB data in the methods section. It is better to show the spatial plot along with the CPCB and US Embassy observations as a scatter. It will show if the model was able to capture the spatial variation in observed $PM_{2.5}$.
How does MERRA-2 compare with the CPCB observations?
Lines 292-293: I do not completely agree with the explanation of transported dust from the Middle East. Is PM10 high over Delhi? Looking at the CALIPSO profile data shown in Beig et al., 2019, it is polluted dust, which is different from desert dust. The authors can also look into the MISR data for dust AOD. The authors have made a statement in section 3.6 that sensitivity tests do not show a major influence of dust being transported from the Middle East.

3.   Have the authors looked into the PBLH from the model and compared with the observations? You might have to derive the PBLH from radiosonde observations. The authors attribute the low $PM_{2.5}$ on Nov 8-10 to the plume rise in the model. My understanding is half of the fire emissions will be released within PBLH and the rest above it. The model is simulating higher PBLH as seen in Fig. 13. I would suggest instead of comparing at the US Embassy only, include observation data from all CPCB stations. PBLH in Delhi during November is less than 1000 m (Nakoudi et al., 2019, AMT). The days when PBLH was low (less than 1000 m) in the model (Nov 7, 11-13), the simulated $PM_{2.5}$ was comparable to the observations.

4.   According to me, sections 3.3, 3.5 and, 3.6 do not add anything new to the paper. The inclusion of missing fire emissions is an important part of the simulation. Also, it is worth to include a comparison with the Kumar et al., 2020 study.

Specific Comments:

1.   Line 33: Ghude et al., 2016 do not mention the long-term health impacts due to an increase in emissions based on current policies.
2.   Lines 42-45: David et al., 2019 show the impact of both transport and emissions on $PM_{2.5}$ in different regions in India. The authors can add results from the study.
3.   Line 78: Add reference - Kumar et al., 2020 (JGR)
4.   Lines 128-129: Studies by Conibear et al., 2018, Venkataraman et al., 2018, and David et al., 2019 have shown that residential energy use is the main source of $PM_{2.5}$ in India.
5.   Lines 203-204: "Irrespective …" From where did the authors get this information?
6.   Multiple places the authors mention "the model was able to capture ..." (For example, Lines 230, 266, 343) – please support your statements with statistics (MB, RMSE).
7.   Replace provinces with states (Lines 311, 315).
8.   Line 374: Change "Fig. 1" to "Fig. 8".
9.   Line 452: What are some of the other meteorological phenomena?
10.  Table 2: Arrange the table as explained in the text (section 2.2).

---

## Referee Comment (RC2) · Anonymous Referee #2 · 6 Oct 2020

General summary

This study investigates the processes causing severe air pollution episodes in New Delhi, India by focusing on one such event observed during November 2017. Specifically, the authors evaluate the impact of biomass burning emissions, long-range transport of dust, and dust emissions on WRF-Chem simulated PM2.5. The model captured the day to day variability but missed the peak pollution peak during 7-10 Nov. Secondary Inorganic Aerosols and Secondary Organic Aerosols are estimated to contribute 30% and 27% of total PM2.5 concentrations in Delhi. Back trajectories showed influence of agricultural fires in Punjab on PM2.5 in Delhi. Long-range transport of dust

is not found to affect air quality in Delhi during this time. High biases in model AOD were observed over central India and low biases over the eastern IGP.

While such studies are very important as they provide important information about the sources leading to dangerous air pollution episodes and inform the mitigation strategies, unfortunately this study does not consider all the key sources of uncertainties in the model simulations and may misinform the mitigation strategies. I am particularly concerned about the ignorance of anthropogenic emission uncertainties that were left out irrespective of several evidences pointing to their key role in the analysis presented in the paper itself. The authors should also provide a clear description of the rationale behind selecting biomass burning and dust aerosols as the most important sources of uncertainties in the model simulations. Below I provide my major and minor comments.

Major comments:

Figure 3 shows that increasing the fire emissions by a factor of 7 is too high and leads to large overestimation of AOD especially in the western part of the domain. Large underestimation in the IGP is reflecting the underestimation of anthropogenic emissions but no sensitivity experiment was designed to look into that. So, the "base" configuration might be showing good performance in Delhi for wrong reasons. Fig. 4a shows an AOD of 4 which is unrealistic for Jaipur. It looks like the authors paid all the attention to getting PM2.5 in Delhi correct simply by upscaling the emissions in the upwind regions but no care was taken to maintain the model performance in the upwind regions. Consequently, the model shows a positive bias in PM2.5 in Punjab with a spatial variability (reflected by standard deviation in Table 3) that is nearly 3.5 times higher than the observed variability in Punjab. Nov. 24 case (no fire day) also supports the idea that the anthropogenic emissions are substantially underestimated. Figure S3 shows that PM2.5 concentrations in Punjab were lower than those in Haryana and increasing the fire emissions by a factor of 7 introduced large uncertainties in model simulations as the model PM2.5 in Punjab became nearly a factor of 4 higher than the observations. If crop residue burning was the major source of this air pollution episode, one must see

the highest observed concentrations in Punjab followed by Haryana and Delhi. Such a pattern exists in the model but not in the observations reflecting that the increasing fire emissions by a factor of 7 is not a reasonable choice. The authors have used back air trajectory to corroborate their assumption that crop residue burning is the major source but backward trajectories only show that the air masses passed over the fire region before arriving at Delhi and are possibly influenced by the fire emissions but they do not tell that agricultural fires are the main source of PM2.5 during this episode. Backward trajectory analysis in Figure 7 also shows that PM2.5 during the pollution episode was driven by a combination of both the anthropogenic and fire emissions. Thus, this approach presents the danger of attributing missing anthropogenic sources to fire sources and may misinform the mitigation strategies if used for that purpose. Therefore, I recommend the authors to include additional sensitivity simulations exploring the role of anthropogenic emission uncertainties.

Fig 4 and related discussion: In addition to the AOD, could you please evaluate the Angstrom exponent to examine if there any differences in the abundance of fine and coarse mode particles and if the model was able to capture those variations. Can you also plot VIIRS AOD in Figure 4 to see if the satellite observed an AOD of 4 in Jaipur?

Fig 5/Table 3: Could you please add a few panels in Figure 5 showing the evaluation against the CPCB data?

Line 301: I think the model observation comparison for the non-episode periods looks good because of the scale of Figure 5a. A zoom into the figure 5a shows that on several occasions, the model showed a bias of up to 100 ug/m3 even in the non-episode period.

Line 308: Are you referring to the model biases relative to MERRA-2 here? If yes, is it reasonable to do so given large biases in MERRA-2 simulated PM2.5 itself as shown in Figure 5a?

Line 368-369: Why do you attribute this error only to transport and not to uncertainties in anthropogenic emissions or other physical processes in the model.

Figure 8: Could you please add PBL height to these panels to help understand whether the smoke was injected in the free troposphere.

Minor comments:

Line 100: Replace '*' with the 'x' and also elsewhere in the paper where you describe the resolution.

Line 194-195: Have you applied any filtering criteria to the CPCB data?

Equation (1): I assume this equation is used to calculate MERRA-2 PM2.5 and not WRF-Chem.

Line 288-289: But the underestimation could also be because of the underestimation of emissions from Delhi.

Line 322-333: This is not true as EDGAR-HTAP provides monthly varying emissions with higher emissions in winter.

Figure 6: It would be useful to mark period 1 and period 2 in the figure.

Line 365: Change "lower" to "smaller".

Line 567: change "intensify" to "accuracy".

---

## Referee Comment (RC3) · Anonymous Referee #3 · 12 Oct 2020

In this study, the authors used the WRF-Chem to simulate the pollution episode during Nov 2017 over New Delhi and evaluated the impacts of biomass burning emissions, long-range transport of dust, and dust emissions on simulated PM2.5. The model was evalated by comparing simulated meteorological parameters and simulated AOD and Pm2.5 with MERRA2 data and observational data. This study provides information on the sources that contribute to the severe PM pollution during Nov 2017. The paper is well organized but improvements in presentation are needed. My comments are as follows. 1. In the design of the simulations, why increasing the emissions by 5, 7 or 10 times? Are these numbers chosen only to get a better simulation of PM in Delhi? 2. Why this study chose to evaluate the impacts from only biomass buring and dust? How

about other anthropogenic emissions which is also important source to severe PM2.5 events. 3. This study simulates the haze event during Nov 2017 by adjusting boundary conditions and emissions, how about other haze evenets in India? How to apply the findings of this study in the simulation of other haze events in India?
* * *

---

## Author Comment (AC1) · 16 Nov 2020

We thank the reviewers for their time and insightful comments, which have substantially improved the manuscript. We have revised the manuscript and addressed the comments raised by the reviewers. The reviewers raised important comments on the rationales for our hypothesis, and the effects of our findings on future simulations. The main purpose of the study is to investigate the sensitivity of model predictions to the main inputs into the model. We apply different scenarios to evaluate the importance of major sources during the November 2017 extreme pollution episode over northern India. We feel this evaluation of inputs is needed to understand the extent that the forward model can be configured to capture the events. A contemporary way to try to capture such events in prediction mode is to employ data assimilation. The data assimilation results compensate for deficiencies in the inputs as well as structural problems within the models. But the effectiveness of data assimilation improves as the capabilities of the forward model improves. Therefore, our results are also important for those using data assimilation to improve predictability. Below, please find our responses to the reviewer's comments. The reviewer's comments are shown in black, our responses are shown in red, and the modified section of the manuscript is shown in blue.

We appreciate your time and comments and look forward to your decision.

Best Regards,

Behrooz Roozitalab, on behalf of all co-authors

**RC1:**

In this study, the authors have used the WRF-Chem to simulate the intensive pollution episode in the Indo-Gangetic Plain (IGP) in November 2017. They carried out 14 sensitivity simulations for different scenarios based on biomass burning emissions, chemical boundary conditions, and dust emissions. The model (base scenario) was evaluated for meteorological parameters (10 m wind speed and direction, 2 m temperature, and surface water vapor) with MERRA-2. The simulated AOD and $PM_{2.5}$ were compared with observations from AERONET and CPCB/US Embassy monitors, respectively. The authors have also looked into the daytime variation in ozone. The study is interesting because the authors try to simulate the $PM_{2.5}$ during November 6-13 using the emissions and aerosol-radiation feedbacks but no assimilation. This study is similar to a recent study published in JGR by Kumar et al., 2020 (https://doi.org/ 10.1029/2020JD033019). Overall, the manuscript needs a major revision. There are too many figures, which makes it hard to get the message across the reader. The labels in the figures are difficult to read. Here are my comments:

**Authors Response:**

**We appreciate the reviewer for thoughtful comments. Please find our responses below. We also moved some of the figures to the supplementary documents and improved the labels quality.**

**Main comments:**

**RC1-1:** I do not understand the hypothesis behind 14 simulations and still not being able to simulate the aerosols. The authors consider FINN_VIIRS_7Xperiod2 (base scenario) as the best scenario but the bias is still high (for AOD and PM2.5) compared to the observations.

**Authors Response:**

**We appreciate the reviewers concerns and we try to clarify them in the followings. (We split the comments and address each part individually)**

**The main purpose of the study is to investigate the sensitivity of model predictions to the main inputs into the model. Here we take the approach of systematically exploring the impacts of different boundary conditions, dust, fire and anthropogenic emissions on the predictions of the pollution episode in November 2017. Based on literature (e.g. (Beig et al., 2019)), three major sources can play a role in this episode: - long-range transported dust incoming from boundaries, - long-range transported dust emitted inside boundary, and – agricultural fires on north-west India. We apply different scenarios to evaluate the importance of mentioned sources for the specific Nov. 2017 episode. We feel this evaluation of inputs is needed to understand the extent that the forward model can be configured to capture the events. A contemporary way to try to capture such events in prediction mode is to employ data assimilation. The data assimilation results compensate for deficiencies in the inputs as well as structural problems within the models. But the effectiveness of data assimilation improves as the capabilities of the forward model improves. Therefore, our results are also important for those using data assimilation to improve predictability. We clarified these points in the revised paper.**

**Text:**

The main purpose of this study is to investigate the sensitivity of model predictions to the main inputs into the model. Prediction of extreme pollution events is important as they have major impacts on people and also make a strong impression regarding the capabilities of models. However, extreme events are hard to predict because they are often heavily impacted by episodic emission sources. Here we take the approach of systematically exploring the impacts of different boundary conditions, dust, fire and anthropogenic emissions on the predictions of the pollution episode in November 2017. A contemporary way to try to capture such events in prediction models is to employ data assimilation (Kumar et al., 2020). The data assimilation results compensate for deficiencies in the inputs as well as structural problems within the models. But the effectiveness of data assimilation improves as the capabilities of the forward model improves. Therefore, our results are also important for those using data assimilation to improve predictability.

**There are different variations of inputs that can be used for each of these sources. In particular:**

1. **We investigate four global data (MOZART, CAMChem, CAMS, and MERRA-2) to find how long-range transported dust incoming from boundaries affected air quality in India.**
2. **We modified the speciation of dust in dust emission module in the model to understand how in-boundary dusts played role in this episode.**
3. **We looked at two different biomass burning (B.B.) inventory, representative of two different methods of biomass burning emission inventories i.e. FRP and burned area, to find better B.B. emission inventory for high resolution modeling of agricultural fires.**
4. **We also did some experiments to reveal whether B.B. emission inventories are either systematically or occasionally biased low.**
5. **We also add one more experiment in the revised version to understand the impacts of anthropogenic emissions as suggested by reviewer 2.**

**As a result, the large number of different scenarios were inevitable. Regarding the comment on high uncertainty in the base scenario after many experiments, we acknowledge the reviewer's concern. However, these experiments document the extent to which modifying these inputs can improve the prediction for this event.**

**For example, we found that statistics improved when we switched from a default scenario (ID: FINN_MERRA2) to the base scenario (ID: FINN_VIIRS_7Xperiod2) as shown below, Fig. S2, and discussion of Fig. 12a. For example, NMB were decreased by 62%. Moreover, the results for the base scenario show a fair performance compared to suggested benchmark criteria by (Emery et al., 2017). On the other hand, Fig. 12b and table S3 depict the statistics after excluding pollution episode days and show even better performance and satisfies "the goal criteria" based on Emery et al. (2017).**

*Table 1 Statistics before (FINN_MERRA2) and after (FINN_VIIRS_7Xperiod2) modifying biomass burning emission inventory*

| Scenario | RMSE ($\mu gm^{-3}$) | NMB (%) | MB (%) |
|---|---|---|---|
| FINN_MERRA2 | 167.88 | -44.32 | -113.14 |
| FINN_VIIRS_7Xperiod2 | 118.47 | -16.6 | -42.38 |

Furthermore, the aspect we looked to improve the modeling results is very different from optimization aspects. As a result, we did not expect to see as good as data assimilation results that strongly constrain the model. We evaluated our results in contrast with a new recent study by Kumar et al. (2020). They used data assimilation to look at a different time-period (while covering the same pollution episode). After assimilating MODIS AOD, their model performance for PM$_{2.5}$ improved significantly for the first day forecast: Mean bias (from -98.7 to -13.7 vs -42.38 in our study) and RMSE (from 167.4 to 117.3 vs 118.47 in our study); that study has better Mean Bias but RMSE values are very close to our values. We have added this evaluation in the revised paper.

Text:

Kumar et al. (2020) assimilated MODIS AOD to WRF-Chem in order to improve the air quality forecasts over Delhi. In their study, Mean Bias for first-day forecast of PM$_{2.5}$ concentration decreased from -98.7 μgm$^{-3}$ to -13.7 μgm$^{-3}$. They also showed that RMSE decreased from 167.4 μgm$^{-3}$ to 117.3 μgm$^{-3}$. Our results from the base scenario (Mean Bias: -42.38 μgm$^{-3}$ and RMSE: 118.47 μgm$^{-3}$) shows comparable results to the data assimilation technique, while still both models are biased low.

RC1-2: From Fig. 3, the simulated AOD is underestimated over the IGP and overestimated over the rest of India. Comparison with AERONET (Fig. 4) shows that MERRA-2 does better over Jaipur because from Fig. 3b it is evident that AOD over Jaipur is in the range 0.5-0.8. Both WRF-Chem and MERRA-2 should have the same resolution while comparing (Fig. S2) and color bar scale comparable to Fig. 3. In conclusion, I think AOD is better simulated by MERRA-2 at both Jaipur and Kanpur. Please include the statistics for AERONET vs WRFChem and MERRA-2 AOD.
Authors Response:

Thanks for the comment. We completely agree that model is biased low over the IGP and biased high elsewhere and we have exclusively mentioned this point in the revised version. This behavior has been also shown in another new study by Jena et al. (2020): Fig. 4 in Jena et al. (2020), where they looked at December 2017 and January 2018.

Regarding the comparison of AOD over Jaipur, we agree that Fig. 3b shows averaged AOD over Jaipur for November was in the range 0.5-0.8, as the reviewer mentioned, and MERRA-2 results are closer to AERONET for non-episode days (AERONET data is missing for episode days). That is reasonable as MERRA-2 modeling system assimilates satellite AOD. VIIRS data also supports high AOD bias of model in Jaipur. However, it is also important that AERONET is missing data for the pollution episode between Nov. 6$^{th}$ and Nov. 13$^{th}$ as shown in Fig.4a. It suggests, as one possibility, that PM concentrations were too high during this period that the instrument was not able to retrieve data at that specific location. In the revised paper, we also include results where we scaled the particles anthropogenic emissions by a factor of two based on some new emission estimates (ID: Base_Anth2X). Using these anthropogenic emissions, averaged AOD bias for the IGP was reduced. This shows the need for improved estimates of biomass burning as well as anthropogenic emissions.

[Figure]

*Figure 1 Bias of AOD at 550nm averaged over November 2017 base on a) base scenario b) base scenario with 2 times more anthropogenic particle emissions (ID: Base_Anth2X)*

**In the revised version, we also added VIIRS retrievals, which supports better performance of MERRA-2. Normalized Mean Bias between the model and AERONET was mentioned in the manuscript (Jaipur: +29.9% and Kanpur: -27.4%) but we added the same metric values between MERRA-2 and AERONET (Jaipur: -20.1% and Kanpur: -1.3%), which supports better performance of MERRA-2 in estimating AOD.**

**We thank the reviewer for the comments on Fig. S2. We modified the map in the revised document.**

**Text:**

Figure 4 shows time series of modeled, MERRA-2 product, VIIRS retrievals, and observed AOD at the AERONET stations, located on Fig.1. AOD values at Kanpur, a station in the eastern IGP, were more than 1.0 before the pollution episode and reached up to 2.0 during the episode days, and decreased to values between 0.5 and 1 for the rest of days. The model captured the general trend although missed high AOD's between Nov. 9[th] and 13[th], while MERRA-2 successfully captured the AOD trend through the whole month, including days with enhanced AOD values. This shows that AOD assimilation in MERRA-2 significantly improves AOD predictions. At Jaipur, located in southern IGP, the model overestimated AOD for the first five days of November. During the pollution episode days, the model is biased high compared to MERRA-2 and VIIRS retrievals. AERONET data showed low AOD values before the pollution episode but did not report values during the pollution episode. It suggests, as one possibility, that PM concentrations were too high during this period that the instrument was not able to retrieve data. After the pollution period, AOD values were lower than 0.5, showing relatively low PM concentrations. In general, MERRA-2 showed better performance in terms of NMB (Kanpur: -1.3% and Jaipur:-20.1%) compared with our model (Kanpur:-27.4% and Jaipur: +29.9%). Comparing averaged AOD with VIIRS retrievals for BASE_ANTHRO2X scenario showed lower bias over the IGP (Fig. in the supplementary document). These results show the need for improved estimates of biomass burning as well as anthropogenic emissions.

[Figure]

*Figure 2 Figure 4 Time series of modeled (green line), VIIRS retrievals (blue triangle), MERRA-2 (red line), and AERONET (black dots) AOD at 550 nm during Nov. 2017 at a) Jaipur, b) Kanpur.*

**RC1-3:  Lines 254-255: There are no major fires during November over western India/Rajasthan (as seen in Fig. 10). Also, there is no major dust event but there is a possibility of anthropogenic dust some of it being unique to the Indian region.**
**Authors Response:**

**Thanks for the comment that reveals the sentence was not clear. By western India for major fires, we meant western IGP and specifically Punjab, as it is clear on one-day fire map for Nov. 5th (Fig. 10).**

**Regarding dust, we have major dust emissions in eastern Pakistan near India, where the PM$_{10}$ concentrations are more than 300 $\mu$gm$^{-3}$, and it can be seen on Fig. 14. However, neither these dust emissions nor long-range transported dust affect Delhi's air quality as explained in sections 3.5 and 3.6. Furthermore, it should be noted that the anthropogenic emissions used in the study include anthropogenic dust.**

**We have modified line254-255 sentence to clarify these points in the revised version.**

**Text:**

Moreover, AODs were high over western IGP, close to major fires of Punjab, with a gradual gradient towards eastern and central India. Dust emission sources in the border of Pakistan also led to high AODs although they did not affect Delhi as discussed in the supplementary document.

**RC1-4:  Lines 270-271: I do not agree with the authors' explanation. As seen in Fig. 3, WRF-Chem is simulating higher AOD values over western India/Rajasthan. The figures do not completely agree with the statements made by the authors.**
**Authors Response:**

**We thank the reviewer and rewrote the paragraph on AERONET data as shown above.**

**RC1-5:  Why are the authors comparing the diurnal variation (Fig. 5a)? Do the emissions have a diurnal variation in the model?**

**Authors Response:**

Thanks for the comment. In this study, all biomass burning emissions have diurnal variation in contrast with monthly anthropogenic emissions. As a result, $PM_{2.5}$ concentrations are subject to both daily atmospheric processes and emissions.

**RC1-6: Why wasn't the $PM_{2.5}$ data from CPCB stations used in Fig. 5a?**

**Authors Response:**

Regarding the question about comparing model results with CPCB stations, we compared our results against CPCB stations in Delhi in Fig. 6. In general during the whole paper, we show the results only at one station (i.e. US Embassy) when we look at time-series and we show daily box and whisker plots when we look at all CPCB stations. Our rationale is that: First, averaging data for time-series may remove some information by smoothing data. Second, we intended to show results from MERRA-2 in our time-series plots. Because of lower resolution of MERRA-2 data, almost all measurement stations in Delhi are located in only one grid cell of MERRA-2, which leads to misinformation. As a result, we decided to show time-series for only US Embassy location and box and whisker plots for CPCB stations. Nonetheless, we show timeseries for US-embassy and all CPCB stations in Delhi below, which shows CPCB averaged values have a similar trend to US-embassy data but with lower peaks.

[Figure]

*Figure 3 $PM_{2.5}$ timeseries in Delhi based on CPCB and US embassy data*

**RC1-7: Fig. 6 includes data from all the CPCB stations, how was the quality check performed on the CPCB data? Please add the details on the quality check of CPCB data in the methods section.**

**Authors Response:**

Thanks for the comment. We didn't apply any filter to this data as we relied on quality control done by CPCB (https://cpcb.nic.in/quality-assurance-quality-control/). However, we studied how applying the following filters, done by Jena et al. (2020) and Kumar et al. (2020), change the dataset consisting of total 12768 hourly data points:

Filter 1: Remove less than 10 µgm$^{-3}$ instances: removes 31 data-points

**Filter 2: Remove the hourly difference between 100 (or 150 or 200) $\mu gm^{-3}$ : removes 186 (or 71 or 31 ) hourly-data**

**Filter 3: Remove values more than 200 (400) $\mu gm^{-3}$ right after NAN value: 33 (19). It basically removes data for Nov. 9th as it was applied after filter #2.**

**We found that the order of applying these filters is important. Below, statistics and timeseries for different orders of filters are presented. Order of filters (1,2,3) removes data for Nov. 9th and significantly improves the model performance over Delhi. We added these findings in the supplementary document and described in the revised version.**

**Text:**

No additional quality control filters, other than the ones by CPCB (https://cpcb.nic.in/quality-assurance-quality-control/), were applied. We evaluated the results after applying the filters proposed by other studies (e.g. Kumar et al. (2020)); they had slight impacts on statistics (shown in the supplementary document).

*Table 2 Effect of applying filters to CPCB data on PM$_{2.5}$ statistics in Delhi*

| Province | Hourly Obs. Mean (±std) ($\mu gm^{-3}$) | Hourly Model Mean (±std) ($\mu gm^{-3}$) | 24-hours NMB (%) | 24-hours NME (%) | 24-hours R (%) |
|---|---|---|---|---|---|
| CPCB-Delhi | 255.5 (±146.6) | 213.9 (±113.9) | -16.6 | 27.6 | 0.48 |
| Only filter 3 | 248.4 (±140.3) | 214.5 (±114.5) | -13.9 | 26.4 | 0.49 |
| Filter123 | 215.5 (±95.5) | 214.8 (±115.2) | -1.9 | 23.6 | 0.64 |
| Filter132 | 248.6 (±140.8) | 214.6 (±114.5) | -13.9 | 26.4 | 0.49 |

[Figure]

*Figure 4 Effect of applying additional filters to CPCB data on averaged PM$_{2.5}$ timeseries in Delhi*

**RC1-8:** **It is better to show the spatial plot along with the CPCB and US Embassy observations as a scatter. It will show if the model was able to capture the spatial variation in observed PM$_{2.5}$. How does MERRA-2 compare with the CPCB observations?**

**Authors Response:**

**As the model resolution is 15km in this study, we usually see more than one measured CPCB stations are located in one model grid cell. For example, 17 stations in Delhi are located in only 6 grid cells; repetition affects the scatter plot (Fig. 5a below). We also observe lower variability in box and whisker plots of the model compared to observation data due to same reason. In other words, scatter plots will not provide enough insights when considering all individual stations. To show the spatial performance of the model, we plot the scatter plot for averaged concentration of different states. Below, we show the scatter plot for Delhi, Haryana, and Rajasthan, which reveals the good spatial performance of the model (Fig. 5b below). Adding data from Punjab to this plot (Fig. 5c below) ,significantly degrades the performance. The reason is due to extremely high bias in Punjab data. Punjab observation data seem to be very uncertain, as it doesn't show any signal of the pollution episode while satellite data show huge amount of agricultural fires during those days. We added the below scatter plots in the supplementary document. However, we think scatter plots were better tools if we had more spatial data (e.g. a gridded dataset).**

**Regarding MERRA-2, due to large grid cell size of MERRA-2 (0.625x0.5deg), all CPCB stations of Delhi are in the same grid cell. As a result, we only look at one representative station (i.e. US Embassy station) when using MERRA-2 data.**

[Figure]

*Figure 5 Scatter plots for a) all stations in Delhi combined b) averaged concentrations in Delhi, Haryana, and Rajasthan c) averaged concentrations in Delhi, Haryana, Rajasthan, and Punjab. Filters are applied to CPCB data.*

**RC1-9: Lines 292-293: I do not completely agree with the explanation of transported dust from the Middle East. Is PM$_{10}$ high over Delhi? Looking at the CALIPSO profile data shown in Beig et al., 2019, it is polluted dust, which is different from desert dust. The authors can also look into the MISR data for dust AOD. The authors have made a statement in section 3.6 that sensitivity tests do not show a major influence of dust being transported from the Middle East.**

**Authors Response:**

**We completely agree with the reviewer and sorry for confusion. Our analysis do not show a major influence of long-range transported dust as discussed in sections 3-5 and 3-6. Since lines 292-293, which are in the section we were looking at the whole month and mentioning other studies' views, may be confusing for the reader on our point of view, we removed the following sentence:**

**"We looked at MERRA-2 surface PM$_{2.5}$ concentration data for the study period to explore if dust was a major source."**

**RC1-10: Have the authors looked into the PBLH from the model and compared with the observations? You might have to derive the PBLH from radiosonde observations. The authors attribute the low PM$_{2.5}$ on Nov 8-10 to the plume rise in the model. My understanding is half of the fire emissions will be released within PBLH and the rest above it. The model is simulating higher PBLH as seen in Fig. 13. I would suggest instead of comparing at the US Embassy only, include observation data from all CPCB stations. PBLH in Delhi during November is less than 1000 m (Nakoudi et al., 2019, AMT). The days when PBLH was low (less than 1000 m) in the model (Nov 7, 11-13), the simulated PM$_{2.5}$ was comparable to the observations.**

**Authors Response:**

**We thank the reviewer for this important comment. Unfortunately, there is not any measured PBLH data available to compare the modeling results. On the other hand, estimating PBLH using sensing data is a challenging task (Nakoudi et al., 2019)and needs specific considerations (Wang and Wang,**

2014). As an example, we used specific humidity (and relative humidity) from radiosonde data for Delhi (provided by university of Wyoming at http://weather.uwyo.edu/upperair/sounding.html ) and attributed the height with lowest vertical gradient as the PBLH. As the figure below shows, WRF-Chem diagnostic PBLH shows higher values (~150m) for Nov. 6th 12UTC (17:30 IST). This result shows PBLH overestimation. We agree with the reviewer that lower PBLH could entrain more aerosol and increase concentrations and we have added the impacts of PBLH on aerosol loading and the importance of accurate PBLH in the revised version. We also compared our modeling results to Nakoudi et al. (2019)findings, which show comparable results. Our hourly averaged PBL has higher heights during daytime compared to ones in winter by Nakoudi et al. (2019). We should note that their results do not cover October and November (i.e. post-monsoon).

On the other hand, we added the PBLH line to the cross sections (white line) in Fig. 8. The line is not obvious on 00 UTC times due to very low extent of the PBL. On Nov. 6th-12, we see very low PBL upwind of Delhi and significant amount of smoke above boundary layer. Therefore, these findings accompanied with Fig. 13b supports the argument that the plume rise in the model released the emissions too high or the model did not mix the smoke sown fast enough (Plume rise module does not have a constrain to release half of the emissions below PBLH and the rest above). We have used new figure in the revised version.

However as mentioned earlier, the main purpose of this study is to investigate the sensitivity of model predictions to the main inputs into the model. We believe another study is required to look at the structure of the model and study the effects of different PBL parameterization modules on PBLH, which is beyond the scope of this paper.

Regarding CPCB stations, as discussed earlier, one MERRA-2 grid cell includes all CPCB stations; as a result, including them will not provide any insights. However, we agree with the reviewer's comment that the model showed comparable $PM_{2.5}$ concentrations with lower PBLH as discussed above.

Text:

Increasing emissions also indirectly influenced modeled air quality over Delhi. As our model configuration included feedbacks, absorbing aerosols in the atmosphere (products of fire emissions) decreased the surface solar radiation budget, changed the dynamics of the atmosphere, reduced the Planetary Boundary Layer (PBL) height, and increased aerosol concentrations. In other words, higher PBLH leads to lower concentrations. For example, Murthy et al. (2020)found that $PM_{2.5}$ concentration decreased up to 14 $\mu gm^{-3}$ for 100m increase in PBLH. Figure 13shows the interactions between PBLH and $PM_{2.5}$ concentration at the location of US embassy. By increasing FINN inventory by 7 times, the PBL height decreased by ~50% on Nov. 6th, (compare FINN_VIIRS_7Xperiod2 and FINN_MERRA-2 panels in Fig. 13). As a result, another study is required to compare modeled PBL heights to observed data (e.g. Nakoudi et al., 2019) and study the effects of different PBL parameterization modules on aerosol concentrations.

[Figure]

*Figure 6 The vertical blue lines represent the quantity (left: specific humidity, right: relative humidity) and vertical red lines show the vertical gradients for mentioned parameters. We defined radiosonde diagnosed PBL height as the minimum gradient (dashed green line) and WRF-Chem diagnosed PBLH is shown in dashed black line.*

[Figure]

*Figure 7 Averaged hourly PBLH during November (top panel) and other different seasons (screenshot from Nakoudi et al. 2019)*

[Figure]

*Figure 8 Figure 8 Vertical cross section of PM₂.₅ concentration through the path shown in Fig. 1 for the days between Nov. 5ᵗʰ and Nov. 10ᵗʰ. For each day, two snapshots are shown at 00UTC (5:30AM local time) and 12UTC (5:30PM local time). The orange star shows the location of Delhi through the path. White line shows the PBL height across the path*

**RC1-11:  According to me, sections 3.3, 3.5 and, 3.6 do not add anything new to the paper. The inclusion of missing fire emissions is an important part of the simulation. Also, it is worth to include a comparison with the Kumar et al., 2020 study.**

**Authors Response:**

**We thank the reviewer for the comment. We moved sections 3-5 and 3-6 to the supplementary document and replace them with one brief discussion at the end of section 3-4 (sensitivities on biomass burning). These two sections show why long-range transported dust both outside and inside of the domain did not influence air quality in Delhi during pollution episode of November 2017. However, we decided to keep section 3-3 (PM speciation) as it conveys important information on secondary aerosols.**

**We appreciate the reviewer for introducing the paper by Kumar et al., 2020. We compared our results to their findings as discussed earlier.**

**Specific Comments:**

**RC1-12:  Line 33: Ghude et al., 2016 do not mention the long-term health impacts due to an increase in emissions based on current policies.**

**Authors Response:**

**We thank the reviewer for catching this mistake. We removed that reference.**

**RC1-13:  Lines 42-45: David et al., 2019 show the impact of both transport and emissions on PM2.5 in different regions in India. The authors can add results from the study.**

**Authors Response:**

**We added David et al. (2019)findings to the revised version.**

**Text:**

Studies show that ozone and particulate matter with diameter less than 2.5 micron (PM$_{2.5}$), are attributed to more than one million individual premature deaths in India (Cohen et al., 2017;HEI, 2018). David et al. (2019) found anthropogenic emission within India led to about 80% of the total premature death due to PM$_{2.5}$ in India. Furthermore as industrial activities are growing, emissions are increasing too; health impacts attributed to long-term exposure air pollution are predicted to increase based on current policies (Conibear et al., 2018a).

**Text:**

David et al. (2019) attributed about 16% of total premature PM$_{2.5}$-related death to emissions outside India.

**RC1-14:  Line 78: Add reference - Kumar et al., 2020 (JGR)**

**Authors Response:**

**We added this reference.**

**RC1-15:  Lines 128-129: Studies by Conibear et al., 2018, Venkataraman et al., 2018, and David et al., 2019 have shown that residential energy use is the main source of PM2.5 in India.**

**Authors Response:**

**Thanks. We emphasized the reviewer's point in the revised version.**

**Text:**

Biomass and biofuel use in residential sector for heating and cooking purposes have significant contributions to air quality in India (Conibear et al., 2018a;David et al., 2019;Venkataraman et al., 2018)

**RC1-16:  Lines 203-204: "Irrespective …" From where did the authors get this information?**

**Authors Response:**

**Thanks for the comment. This was from personal inspections but to be clear we modified the sentence.**

**Text:**

Irrespective of this condition, stations are placed on top of the buildings with restricted clean flow of air (personal inspections).

**RC1-17:** Multiple places the authors mention "the model was able to capture …" (For example, Lines 230, 266, 343) – please support your statements with statistics (MB, RMSE).

**Authors Response:**

**Thanks. We added statistics when it was not mentioned.**

**RC1-18:** Replace provinces with states (Lines 311, 315).

**Authors Response:**

**Thanks. We replaced all 'provinces' with 'states'**

**RC1-19:** Line 374: Change "Fig. 1" to "Fig. 8".

**Authors Response:**

**Thanks. We clarified it.**

**RC1-20:** Line 452: What are some of the other meteorological phenomena?

**Authors Response:**

**Thanks for the comment. We mostly meant thick fires can be identified as clouds in retrieval algorithm or they may be actual clouds and large amount of water vapors, leading to biases. Bright surfaces (in deserts) are other uncertainty sources. We modified the sentence:**

**Text:**

Some studies have shown that thick fires can be identified as clouds in retrieval algorithms (Dekker et al., 2019;Huijnen et al., 2016).

**RC1-21:** Table 2: Arrange the table as explained in the text (section 2.2).

**Authors Response:**

**Thanks for the comment. We rearranged it to first show the experiments on anthropogenic and biomass burning emissions, then on boundary conditions, and lastly on dust emissions, to follow section 2.2. We had to add the new sensitivity test to the bottom of the table as to keep the format.**

**References**

Beig, G., Srinivas, R., Parkhi, N. S., Carmichael, G., Singh, S., Sahu, S. K., Rathod, A., and Maji, S.: Anatomy of the winter 2017 air quality emergency in Delhi, Science of The Total Environment, 681, 305-311, 2019.

David, L. M., Ravishankara, A., Kodros, J. K., Pierce, J. R., Venkataraman, C., and Sadavarte, P.: Premature mortality due to PM2. 5 over India: Effect of atmospheric transport and anthropogenic emissions, GeoHealth, 3, 2-10, 2019.

Dekker, I. N., Houweling, S., Pandey, S., Krol, M., Röckmann, T., Borsdorff, T., Landgraf, J., and Aben, I.: What caused the extreme CO concentrations during the 2017 high-pollution episode in India?, Atmospheric Chemistry and Physics, 19, 3433-3445, 2019.

Emery, C., Liu, Z., Russell, A. G., Odman, M. T., Yarwood, G., and Kumar, N.: Recommendations on statistics and benchmarks to assess photochemical model performance, Journal of the Air & Waste Management Association, 67, 582-598, 2017.

Huijnen, V., Wooster, M. J., Kaiser, J. W., Gaveau, D. L., Flemming, J., Parrington, M., Inness, A., Murdiyarso, D., Main, B., and Van Weele, M.: Fire carbon emissions over maritime southeast Asia in 2015 largest since 1997, Scientific reports, 6, 26886, 2016.

Jena, C., Ghude, S. D., Kulkarni, R., Debnath, S., Kumar, R., Soni, V. K., Acharja, P., Kulkarni, S. H., Khare, M., and Kaginalkar, A. J.: Evaluating the sensitivity of fine particulate matter (PM 2.5) simulations to chemical mechanism in Delhi, Atmospheric Chemistry and Physics Discussions, 1-28, 2020.

Kumar, R., Ghude, S. D., Biswas, M., Jena, C., Alessandrini, S., Debnath, S., Kulkarni, S., Sperati, S., Soni, V. K., and Nanjundiah, R. S.: Enhancing Accuracy of Air Quality and Temperature Forecasts During Paddy Crop Residue Burning Season in Delhi Via Chemical Data Assimilation, Journal of Geophysical Research: Atmospheres, 125, e2020JD033019, 2020.

Murthy, B., Latha, R., Tiwari, A., Rathod, A., Singh, S., and Beig, G.: Impact of mixing layer height on air quality in winter, Journal of Atmospheric and Solar-Terrestrial Physics, 197, 105157, 2020.

Nakoudi, K., Giannakaki, E., Dandou, A., Tombrou, M., and Komppula, M.: Planetary boundary layer height by means of lidar and numerical simulations over New Delhi, India, Atmospheric Measurement Techniques, 12, 2019.

Wang, X., and Wang, K.: Estimation of atmospheric mixing layer height from radiosonde data, Atmospheric Measurement Techniques Discussions, 7, 2014.

---

## Author Comment (AC2) · 16 Nov 2020

We thank the reviewers for their time and insightful comments, which have substantially improved the manuscript. We have revised the manuscript and addressed the comments raised by the reviewers. The reviewers raised important comments on the rationales for our hypothesis, and the effects of our findings on future simulations. The main purpose of the study is to investigate the sensitivity of model predictions to the main inputs into the model. We apply different scenarios to evaluate the importance of major sources during the November 2017 extreme pollution episode over northern India. We feel this evaluation of inputs is needed to understand the extent that the forward model can be configured to capture the events. A contemporary way to try to capture such events in prediction mode is to employ data assimilation. The data assimilation results compensate for deficiencies in the inputs as well as structural problems within the models. But the effectiveness of data assimilation improves as the capabilities of the forward model improves. Therefore, our results are also important for those using data assimilation to improve predictability. Below, please find our responses to the reviewer's comments. The reviewer's comments are shown in black, our responses are shown in red, and the modified section of the manuscript is shown in blue.

We appreciate your time and comments and look forward to your decision.

Best Regards,

Behrooz Roozitalab, on behalf of all co-authors

**RC2**:**

This study investigates the processes causing severe air pollution episodes in New Delhi, India by focusing on one such event observed during November 2017. Specifically, the authors evaluate the impact of biomass burning emissions, long-range transport of dust, and dust emissions on WRF-Chem simulated PM2.5. The model captured the day to day variability but missed the peak pollution peak during 7-10 Nov. Secondary Inorganic Aerosols and Secondary Organic Aerosols are estimated to contribute 30% and 27% of total PM2.5 concentrations in Delhi. Back trajectories showed influence of agricultural fires in Punjab on PM2.5 in Delhi. Long-range transport of dust is not found to affect air quality in Delhi during this time. High biases in model AOD were observed over central India and low biases over the eastern IGP.

While such studies are very important as they provide important information about the sources leading to dangerous air pollution episodes and inform the mitigation strategies, unfortunately this study does not consider all the key sources of uncertainties in the model simulations and may misinform the mitigation strategies. I am particularly concerned about the ignorance of anthropogenic emission uncertainties that were left out irrespective of several evidences pointing to their key role in the analysis presented in the paper itself. The authors should also provide a clear description of the rationale behind selecting biomass burning and dust aerosols as the most important sources of uncertainties in the model simulations. Below I provide my major and minor comments. Authors Response:

We appreciate the reviewer for pointing to important issues. We addressed the comments and concerns here and below:

We share the view about the critical role of anthropogenic emissions roles in air quality over the IGP. The uncertainty in anthropogenic emissions lead to concentration biases for typical days. Moreover, we acknowledge the importance of anthropogenic emissions since emissions due to heating also increase as the weather gets cold during Oct. and Nov. We added a scenario in which we increased all particles anthropogenic emissions by a factor of 2 based on recent emission work in Delhi (ID: Base\_Anth2X). In following comments, we present its results.

However, agricultural fires have a more significant contribution in post-monsoon extreme pollution events in Delhi (Kulkarni et al., 2020). Moreover, (Beig et al., 2019) showed that extreme pollution episodes during November 2017 was mainly due to agricultural fires and long-range transported dust. Lines 46-52 discuss these points although we agree that there are some exceptions, too. For example, extensive use of firecrackers and fireworks in the Diwali festival on October 20th in 2017 led to PM2.5 concentrations above 600 (µgm-3) Therefore, we focused only on November to exclude that episode. On the other hand, our simulation results after excluding extreme pollution days show fair statistics (Table S3). We highlighted the importance of anthropogenic emissions in the revised paper and tried to express the reviewers point in the study limitations section in the revised version:

**Text:**

During this study, we did not primarily focus on improving anthropogenic emissions over the region in order to capture extreme pollution episode. However, anthropogenic emissions are low in global emission inventories and needed to be improved (Jat et al., 2020). Moreover, very low biased concentrations for some days and trajectory results suggest the existence of some other sources, primarily anthropogenic sources, upwind of Delhi that should be studied more.

**Main comments:**

RC2-1: Figure 3 shows that increasing the fire emissions by a factor of 7 is too high and leads to large overestimation of AOD especially in the western part of the domain. Large underestimation in the IGP is reflecting the underestimation of anthropogenic emissions but no sensitivity experiment was designed to look into that. So, the "base" configuration might be showing good performance in Delhi for wrong reasons.

**Authors Response:**

We appreciate reviewer's genuine and important concerns. Please find our responses, below. (We split the comments and address each part individually)

Regarding Biases in Fig. 3: As you and reviewer 1 mentioned, we completely agree that model is biased low over the IGP and biased high elsewhere and we have exclusively mentioned this point in the revised version. We acknowledge that uncertainty of anthropogenic emissions is playing an important role in these biases. We did another experiment where we increased anthropogenic emissions for all the particles by a factor of 2 (ID: Base\_Anth2X). This modification increased PM2.5 concentrations in Delhi up to ~150 µgm-3, which led to overestimation (in contrast to underestimation in base scenario) at most of non-episode days (time-series shown below). Although this scenario did not help capturing concentrations during the episode, it confirms the need for better anthropogenic emissions. On the other hand, it increased the AOD bias over southern IGP while reduced the bias over IGP (bias map shown below). These results suggest anthropogenic emission inventories have higher bias over IGP compared with non-IGP regions. However, we acknowledge the importance of having dynamic (daily) anthropogenic emission inventory.

**Text:**

Although different meteorological parameters can be responsible for the biases, accuracy of anthropogenic emissions is important. For example, recent local anthropogenic emission inventories developed for Delhi have higher particle emissions than in the regional inventory used in this study, which impacts modeled  $PM_{2.5}$  concentrations for typical days (Kulkarni et al., 2020). We conducted BASE\_ANTHRO2X scenario to investigate the effect of uncertainties in the anthropogenic emissions. This scenario increased  $PM_{2.5}$  concentrations in Delhi up to ~150 µgm-3, which led to overestimation (in contrast to underestimation in base scenario) at many of non-episode days (Fig. in the supplementary document). Although this scenario did not help in capturing the high concentrations during the episode, it confirms the need for better anthropogenic emissions. On the other hand, it reduced the bias over IGP (Fig. in the supplementary document). These results point out the need for best estimates of emissions of both anthropogenic and biomass.

Figure 1 a) Timeseries for PM2.5 concentration at the location of US embassy using Base scenario and Base\_Anth2X scenario B) Bias of AOD at 550nm averaged over November 2017 base on b) base scenario c) base scenario with 2 times more anthropogenic particle emissions (ID: Base\_Anth2X)

In addition, our experiments were primarily focused to capture the extreme pollution episode over Delhi as the reviewer pointed out. On the other hand, we would like to mention an important point regarding the accuracy of the base scenario for other locations:

Below, we show the AOD biases for our base scenario (as in Fig3.c) on left panel, FINN\_MERRA2 scenario (a scenario without any enhancement on fire emissions) on middle, and the difference between these two scenarios on the right panel.

The bias pattern of FINN\_MERRA2 has also been reported in another study by Jena et al. (2020). They looked at a different time period (Dec. 2017 to Jan. 2018) but they show same pattern with lower values (which is most possibly due to lower concentrations in their period of interest). Their results (specifically Fig.4 in Jena et al., 2020) support the importance of anthropogenic emissions as the reviewer mentioned, and we acknowledge that as discussed above.

On the other hand, looking at base and FINN\_MERRA2 reveals that we clearly improved the AOD results for Punjab. It also shows low bias of FINN\_MERRA2 shifted to high bias of base scenario for Haryana. The difference between base scenario and FINN\_MERRA2 scenario (right panel) shows the impact of increasing FINN emissions by 7 times for a 8-days period; it increased the mean AOD biases over the whole domain by 0.09 (±0.23).

Figure 2 Bias of AOD at 550nm averaged over November 2017 base on a) base scenario b) a scenario without any modifications on biomass burning emissions (ID: FINN\_MERRA2), c) difference between Base and FINN\_MERRA2

RC2-2: Fig. 4a shows an AOD of 4 which is unrealistic for Jaipur. It looks like the authors paid all the attention to getting PM2.5 in Delhi correct simply by upscaling the emissions in the upwind regions but no care was taken to maintain the model performance in the upwind regions. Consequently, the model shows a positive bias in PM2.5 in Punjab with a spatial variability (reflected by standard deviation in Table 3) that is nearly 3.5 times higher than the observed variability in Punjab. Nov. 24 case (no fire day) also supports the idea that the anthropogenic emissions are substantially underestimated.

**Authors Response:**

Regarding Fig4.a, we agree that the model is generally biased high over the Jaipur. Moreover, VIIRS data also show low AOD values for Jaipur during episode days. That is reasonable as MERRA-2 modeling system assimilates satellite AOD. However, it is also important that AERONET is missing data for the pollution episode between Nov. 6th and Nov. 13th as shown in Fig.4a. It suggests, as one possibility, that PM concentrations were too high during this period that the instrument was not able to retrieve data at that specific coordinates. We modified the discussion on Fig.4 in the revised version.

**Text:**

Figure 4 shows time series of modeled, MERRA-2 product, VIIRS retrievals, and observed AOD at the AERONET stations, located on Fig.1. AOD values at Kanpur, a station in the eastern IGP, were more than 1.0 before the pollution episode and reached up to 2.0 during the episode days, and decreased to values between 0.5 and 1 for the rest of days. The model captured the general trend although missed high AOD's between Nov. 9th and 13th, while MERRA-2 successfully captured the AOD trend through the whole month, including days with enhanced AOD values. This shows that AOD assimilation in MERRA-2 significantly improves AOD predictions. At Jaipur, located in southern IGP, the model overestimated AOD for the first five days of November. During the pollution episode days, the model is biased high compared to MERRA-2 and VIIRS retrievals. AERONET data showed low AOD values before the pollution episode but did not report values du

---

## Author Comment (AC3) · 16 Nov 2020

We thank the reviewers for their time and insightful comments, which have substantially improved the manuscript. We have revised the manuscript and addressed the comments raised by the reviewers. The reviewers raised important comments on the rationales for our hypothesis, and the effects of our findings on future simulations. The main purpose of the study is to investigate the sensitivity of model predictions to the main inputs into the model. We apply different scenarios to evaluate the importance of major sources during the November 2017 extreme pollution episode over northern India. We feel this evaluation of inputs is needed to understand the extent that the forward model can be configured to capture the events. A contemporary way to try to capture such events in prediction mode is to employ data assimilation. The data assimilation results compensate for deficiencies in the inputs as well as structural problems within the models. But the effectiveness of data assimilation improves as the capabilities of the forward model improves. Therefore, our results are also important for those using data assimilation to improve predictability. Below, please find our responses to the reviewer's comments. The reviewer's comments are shown in black, our **responses** are shown in **red**, and the modified section of the **manuscript** is shown in **blue**.

We appreciate your time and comments and look forward to your decision.

Best Regards,

Behrooz Roozitalab, on behalf of all co-authors

**RC3:**

**In this study, the authors used the WRF-Chem to simulate the pollution episode during Nov 2017 over New Delhi and evaluated the impacts of biomass burning emissions, long-range transport of dust, and dust emissions on simulated $PM_{2.5}$. The model was evaluated by comparing simulated meteorological parameters and simulated AOD and $PM_{2.5}$ with MERRA2 data and observational data. This study provides information on the sources that contribute to the severe PM pollution during Nov 2017. The paper is well organized but improvements in presentation are needed.**

**Authors Response:**

**We appreciate the reviewer for pointing to important issues. We try to address the comments and concerns here and below:**

**My comments are as follows.**

**RC3-1:  In the design of the simulations, why increasing the emissions by 5, 7 or 10 times? Are these numbers chosen only to get a better simulation of PM in Delhi?**

**Authors Response:**

**We thank the reviewer for the point. Yes, we used simulation results in Delhi as the criteria for choosing the proper scaling factor. Moreover, the increasing factors were chosen arbitrarily and we have mentioned this as the limitation of our study in the revised version. In this study, we intended primarily to show the bias in biomass burning emission inventory is not systematic and clarify high uncertainty for extremely polluted days. Another bias correction study is required to find the relation between highly polluted days and optimized increasing factor to modify biomass burning emission inventories.**

**Text:**

On the other hand, choosing the multiplication factor for increasing fire emissions was arbitrary in this study. Due to scarcity of observation data, we were not able to apply complicated mathematical scaling techniques based on data assimilation to scale the fire emissions (Saide et al., 2015).

**RC3-2:  Why this study chose to evaluate the impacts from only biomass burning and dust? How about other anthropogenic emissions which is also important source to severe $PM_{2.5}$ events.**

**Authors Response:**

**We focused on biomass burning and dust emissions in this study based on previous studies during this period (e.g. Beig et al. (2019)). However, we acknowledge the importance of anthropogenic emissions since emissions due to heating also increase as the weather gets cold during Oct. and Nov. We conducted another scenario, in which we increased the particles anthropogenic emissions by a factor of 2 (ID: BASE_ANTHRO2X). This modification increased $PM_{2.5}$ concentrations in Delhi up to ~150 μgm$^{-3}$, which led to overestimation (in contrast to underestimation in base scenario) at most of non-episode days (time-series shown below). Although this scenario did not help capturing concentrations**

during the episode, it confirms the need for better anthropogenic emissions. On the other hand, it increased the AOD bias over southern IGP while reduced the bias over IGP (bias map shown below). These results suggest anthropogenic emission inventories have higher bias over IGP compared with non-IGP regions. However, we acknowledge the importance of having dynamic (daily) anthropogenic emission inventory.

**Text:**

Although different meteorological parameters can be responsible for the biases, accuracy of anthropogenic emissions is important. For example, recent local anthropogenic emission inventories developed for Delhi have higher particle emissions than in the regional inventory used in this study, which impacts modeled PM$_{2.5}$ concentrations for typical days (Kulkarni et al., 2020). We conducted BASE_ANTHRO2X scenario to investigate the effect of uncertainties in the anthropogenic emissions. This scenario increased PM$_{2.5}$ concentrations in Delhi up to ~150 µgm$^{-3}$, which led to overestimation (in contrast to underestimation in base scenario) at many of non-episode days (Fig. in the supplementary document). Although this scenario did not help in capturing the high concentrations during the episode, it confirms the need for better anthropogenic emissions. On the other hand, it reduced the bias over IGP (Fig. in the supplementary document). These results point out the need for best estimates of emissions of both anthropogenic and biomass.

[Figure]

*Figure 1 a) Timeseries for PM$_{2.5}$ concentration at the location of US embassy using Base scenario and Base_Anth2X scenario B) Bias of AOD at 550nm averaged over November 2017 base on b) base scenario c) base scenario with 2 times more anthropogenic particle emissions (ID: Base_Anth2X)*

**RC3-3:** This study simulates the haze event during Nov 2017 by adjusting boundary conditions and emissions, how about other haze events in India? How to apply the findings of this study in the simulation of other haze events in India?
**Authors Response:**

We appreciate the reviewer for this important point. The main purpose of the study is to investigate the sensitivity of model predictions to the main inputs into the model. Prediction of extreme pollution events is important as they have major impacts on people and also make a strong impression regarding the capabilities of models. However, extreme events are hard to predict because they are often heavily impacted by episodic emission sources. Here we take the approach of systematically exploring the impacts of different boundary conditions, dust, fire and anthropogenic emissions on the predictions of the pollution episode in November 2017. We feel this evaluation of inputs is needed to understand the extent that which the forward model can be configured to capture the events. A contemporary way to try to capture such events in prediction mode is to employ data assimilation. The data assimilation results compensate for deficiencies in the inputs as well as structural problems

**within the models. But the effectiveness of data assimilation improves as the capabilities of the forward model improves. Therefore, our results are also important for those using data assimilation to improve predictability.**

**Below, please also find other findings:**

- **We showed that biomass burning emission inventories miss some small fire emission and introduced a new technique to use satellite data to fill these missing sources.**
- **We showed that biomass burning emission inventories occasionally underestimate emissions in hazy events up to 7 times lower, where bias correction techniques need to be applied.**
- **We showed either the plume rise in the model release the agricultural fire emissions too high or the model does not mix the smoke down fast enough. These should be considered in future hazy event simulations.**
- **We found Secondary aerosols comprise more than half of the particles in Delhi. It suggests simple aerosol modules like GOCART cannot simulate the actual speciation of particles in Delhi.**

**Text:**

The main purpose of this study is to investigate the sensitivity of model predictions to the main inputs into the model. Prediction of extreme pollution events is important as they have major impacts on people and also make a strong impression regarding the capabilities of models. However, extreme events are hard to predict because they are often heavily impacted by episodic emission sources. Here we take the approach of systematically exploring the impacts of different boundary conditions, dust, fire and anthropogenic emissions on the predictions of the pollution episode in November 2017. A contemporary way to try to capture such events in prediction models is to employ data assimilation (Kumar et al., 2020). The data assimilation results compensate for deficiencies in the inputs as well as structural problems within the models. But the effectiveness of data assimilation improves as the capabilities of the forward model improves. Therefore, our results are also important for those using data assimilation to improve predictability.

**References**

Beig, G., Srinivas, R., Parkhi, N. S., Carmichael, G., Singh, S., Sahu, S. K., Rathod, A., and Maji, S.: Anatomy of the winter 2017 air quality emergency in Delhi, Science of The Total Environment, 681, 305-311, 2019.

Kulkarni, S. H., Ghude, S. D., Jena, C., Karumuri, R. K., Sinha, B., Sinha, V., Kumar, R., Soni, V., and Khare, M.: How Much Does Large-Scale Crop Residue Burning Affect the Air Quality in Delhi?, Environmental Science & Technology, 54, 4790-4799, 2020.

Kumar, R., Ghude, S. D., Biswas, M., Jena, C., Alessandrini, S., Debnath, S., Kulkarni, S., Sperati, S., Soni, V. K., and Nanjundiah, R. S.: Enhancing Accuracy of Air Quality and Temperature Forecasts During Paddy Crop Residue Burning Season in Delhi Via Chemical Data Assimilation, Journal of Geophysical Research: Atmospheres, 125, e2020JD033019, 2020.

Saide, P. E., Peterson, D. A., da Silva, A., Anderson, B., Ziemba, L. D., Diskin, G., Sachse, G., Hair, J., Butler, C., and Fenn, M.: Revealing important nocturnal and day-to-day variations in fire smoke emissions through a multiplatform inversion, Geophysical Research Letters, 42, 3609-3618, 2015.

---

## Referee Report (RR1)

Review of "Improving regional air quality predictions in the Indo-Gangetic Plain – Case study of an intensive pollution episode in November 2017" by Roozitalab et al. (Paper #acp-2020-744).

The authors have made substantial changes in the revised manuscript and have addressed the reviewers' comments. There is clarity on the rationale of the study and the importance of different sources on extreme pollution episodes in the Indo-Gangetic Plain (IGP). I still have a major comment on the usage and evaluation of the model using CPCB data.

Main comments:

1.      Even though on the CPCB website they mention quality control, the user has to be careful while using CPCB data. I have used CPCB data and I am sure the data is not quality controlled. I am not sure if the authors have spent time looking into the CPCB data. Quite often you will find very low values, abrupt changes in values, and high values after missing data. Most probably the CPCB data from Punjab might not have been quality checked or the filters have not been changed timely and failed to detect the high concentrations. I do not understand the logic for applying filters in the order 1,3, and 2 in the revised manuscript.

2.      The authors show the changes in the statistics after applying the filters but have not implemented it in the revised manuscript. I would suggest using CPCB data after applying filters 1, 2, and 3.

3.      I do not completely agree with the author's explanation about not using $PM_{2.5}$ data from CPCB stations. Using data from one location (US embassy) is not representative of the Delhi region. For MERRA-2, the Delhi region will fall into 4 MERRA-2 grids. The CPCB stations used in the study lie in 2 MERRA-2 grids. The averaged CPCB $PM_{2.5}$ concentrations within each grid can be compared with the MERRA-2 $PM_{2.5}$. Are the authors considering the model data to the nearest CPCB location and averaging for the same hours for which the observational data is available in the box and whisker plot? If more than one CPCB station lies in a WRF-Chem grid, do the authors average CPCB data before comparing it with the model?

Specific Comments:

1.  Line 8: Mention the pollution episode days.
2.  Line 20: "The model AODs were biased high …". Add the MB value.
3.  Line 59: Add India's and WHO's standards for $PM_{2.5}$.
4.  Line 69: Add India's standard for ozone.
5.  Line 112: Correct 0.625º×0.5º to 0.5º×0.625º. MERRA-2 data resolution is 0.5º latitude × 0.625º longitude. Make changes in other places in the manuscript.
6.  Figure 2. Change the 'OBS' legend to 'MERRA-2'. Why wasn't WRF-Chem re-gridded to MERRA-2 resolution for comparison?
7.  Are Figures 2e-2j hourly averaged values for the month of November? Please mention it in the caption.
8.  Line 289-291: "It suggests, …" Avoid making such statements when you aren't sure about it.
9.  Line 294: "These results show the need for improved estimates …". It will be worth discussing the paper by Pan et al, ACP 2020 on the usage of different biomass emissions.
10. Line 338: Replace 'was biased high' with 'simulated high $PM_{2.5}$' and 'biased low' with 'simulated low $PM_{2.5}$'. Figures 5b and 5c show the $PM_{2.5}$ concentrations.

11. Line 363: "On the other hand, it reduced the …" Add the bias increased over the rest of India.